# Molecular basis for substrate specificity of the Phactr1/PP1 phosphatase holoenzyme

Roman O Fedoryshchak[1†], Magdalena Přechová[1†‡], Abbey M Butler[1,2], Rebecca Lee[1,2], Nicola O'Reilly[3], Helen R Flynn[4], Ambrosius P Snijders[4], Noreen Eder[4,5], Sila Ultanir[5], Stephane Mouilleron[2]*, Richard Treisman[1]*

[1]Signalling and Transcription Laboratory, The Francis Crick Institute, London, United Kingdom; [2]Structural Biology Science Technology Platform, The Francis Crick Institute, London, United Kingdom; [3]Peptide Chemistry Science Technology Platform, The Francis Crick Institute, London, United Kingdom; [4]Proteomics Science Technology Platform, The Francis Crick Institute, London, United Kingdom; [5]Kinases and Brain Development Laboratory The Francis Crick Institute, London, United Kingdom

*For correspondence:
stephane.mouilleron@crick.ac.uk
(SM);
Richard.Treisman@Crick.ac.uk (RT)

[†]These authors contributed
equally to this work

Present address: [‡]Laboratory of
Integrative Biology, Institute of
Molecular Genetics of the Czech
Academy of Sciences, Prague,
Czech Republic

Competing interests: The
authors declare that no
competing interests exist.

Reviewing editor: Tony Hunter,
Salk Institute for Biological
Studies, United States

**Abstract** PPP-family phosphatases such as PP1 have little intrinsic specificity. Cofactors can target PP1 to substrates or subcellular locations, but it remains unclear how they might confer sequence-specificity on PP1. The cytoskeletal regulator Phactr1 is a neuronally enriched PP1 cofactor that is controlled by G-actin. Structural analysis showed that Phactr1 binding remodels PP1's hydrophobic groove, creating a new composite surface adjacent to the catalytic site. Using phosphoproteomics, we identified mouse fibroblast and neuronal Phactr1/PP1 substrates, which include cytoskeletal components and regulators. We determined high-resolution structures of Phactr1/PP1 bound to the dephosphorylated forms of its substrates IRSp53 and spectrin αII. Inversion of the phosphate in these holoenzyme-product complexes supports the proposed PPP-family catalytic mechanism. Substrate sequences C-terminal to the dephosphorylation site make intimate contacts with the composite Phactr1/PP1 surface, which are required for efficient dephosphorylation. Sequence specificity explains why Phactr1/PP1 exhibits orders-of-magnitude enhanced reactivity towards its substrates, compared to apo-PP1 or other PP1 holoenzymes.

## Introduction

PPP family phosphatases are metalloenzymes that carry out the majority of protein serine/threonine dephosphorylation (*Brautigan and Shenolikar, 2018*). The three Protein Phosphatase 1 (PP1) iso-forms regulate diverse cellular processes, acting in partnership with over 200 different PP1-interacting proteins (PIPs). Some PIPs are PP1 substrates, but others are PP1 cofactors, which variously determine substrate specificity, subcellular targeting and/or coupling to regulatory pathways (*Bollen et al., 2010*; *Cohen, 2002*). The PP1 catalytic site lies at the intersection of three putative substrate-binding grooves (*Egloff et al., 1995*; *Goldberg et al., 1995*), and PIPs can interact both with these grooves and with other PP1 surface features. To do this they use a variety of short sequence elements, of which the best understood is the RVxF motif (*Choy et al., 2014*; *Egloff et al., 1997*; *Hendrickx et al., 2009*; *Hurley et al., 2007*; *O'Connell et al., 2012*; *Ragusa et al., 2010*; *Terrak et al., 2004*).

Unlike protein Ser/Thr kinases, PP1 exhibits little sequence-specificity by itself (*Brautigan and Shenolikar, 2018*; *Miller and Turk, 2018*). Moreover, no instances of PIP-induced sequence-specificity are known, although it is well established that PIPs can enhance or inhibit PP1 activity towards

**eLife digest** Specific arrangements of atoms such as bulky phosphate groups can change the activity of a protein and how it interacts with other molecules. Enzymes called kinases are responsible for adding these groups onto a protein, while phosphatases remove them.

Kinases are generally specific for a small number of proteins, adding phosphate groups only at sites embedded in a particular sequence in the target protein. Phosphatases, however, are generalists: only a few different types exist, which exhibit little target sequence specificity. Partner proteins can attach to phosphatases to bring the enzymes to specific locations in the cell, or to deliver target proteins to them; yet, it is unclear whether partner binding could also change the structure of the enzyme so the phosphatase can recognise only a restricted set of targets.

To investigate this, Fedoryshchak, Přechová et al. studied a phosphatase called PP1 and its partner, Phactr1. First, the structure of the Phactr1/PP1 complex was examined using biochemistry approaches and X-ray crystallography. This showed that binding of Phactr1 to PP1 creates a new surface pocket, which comprised elements of both proteins. In particular, this composite pocket is located next to the part of the PP1 enzyme responsible for phosphate removal.

Next, mass spectrometry and genetics methods were harnessed to identify and characterise the targets of the Phactr1/PP1 complex. Structural analysis of the proteins most susceptible to Phactr1/PP1 activity showed that they had particular sequences that could interact with Phactr1/PP1's composite pocket. Further experiments revealed that, compared to PP1 acting alone, the pocket increased the binding efficiency and reactivity of the complex 100-fold.

This work demonstrates that a partner protein can make phosphatases more sequence-specific, suggesting that future studies could adopt a similar approach to examine how other enzymes in this family perform their role. In addition, the results suggest that it will be possible to design Phactr1/PP1-specific drugs that act on the composite pocket. This would represent an important proof of principle, since current phosphatase-specific drugs do not target particular phosphatase complexes.

particular substrates (*Ichikawa et al., 1996*; *Johnson et al., 1996*). Some PIPs contain autonomous substrate-binding domains, which facilitate substrate recruitment (*Boudrez et al., 2000*; *Choy et al., 2015*), while others constrain substrate specificity by occluding PP1 surfaces such as the RVxF-binding pocket and/or substrate-binding grooves (*Hirschi et al., 2010*; *Ragusa et al., 2010*). Interestingly, several PIPs extend the PP1 substrate-binding grooves and/or significantly alter PP1 surface electrostatics, without altering the conformation of the catalytic site (*O'Connell et al., 2012*; *Ragusa et al., 2010*; *Terrak et al., 2004*). How this might affect substrate selection remains unclear, as the sequence-specificities of these PIP/PP1 holoenzymes, and how they bind their substrates, have not been characterised.

The four Phosphatase and actin regulator (Phactr) proteins (*Allen et al., 2004*; *Sagara et al., 2003*) are novel PIPs that are implicated in cytoskeletal regulation in animal models (*Hamada et al., 2018*; *Kim et al., 2007*; *Zhang et al., 2012*) and cell culture settings (*Huet et al., 2013*; *Wiezlak et al., 2012*). The Phactrs bind G-actin via multiple RPEL motifs present in their conserved N- and C-terminal regions (*Huet et al., 2013*; *Mouilleron et al., 2012*; *Sagara et al., 2009*; *Wiezlak et al., 2012*). Their C-terminal RPEL domain overlaps the PP1 binding sequence (*Allen et al., 2004*; *Larson et al., 2008*; *Sagara et al., 2003*), and G-actin competes with PP1 for Phactr binding (*Huet et al., 2013*; *Wiezlak et al., 2012*). As a result, like other RPEL proteins, G-actin/Phactr interactions respond to fluctuations in actin dynamics (*Diring et al., 2019*; *Miralles et al., 2003*; *Vartiainen et al., 2007*). Extracellular signals, acting through the 'rho-actin' signal pathway, thus control Phactr/PP1 complex formation by inducing changes in cellular G-actin concentration (*Wiezlak et al., 2012*).

The biochemical function of Phactr/PP1 complexes has been unclear. Phactr1 and Phactr3 inhibit dephosphorylation of phosphorylase a by PP1 in vitro (*Allen et al., 2004*; *Sagara et al., 2003*), but Phactr4/PP1 complex formation is associated with cofilin dephosphorylation in vivo (*Huet et al., 2013*; *Zhang et al., 2012*). Here, we show that Phactr1 confers sequence-specificity on the Phactr1/PP1 holoenzyme. We identify substrates for Phactr1/PP1, and determine the structures of Phactr1/PP1-substrate complexes. We show that efficient catalysis requires interactions between conserved

hydrophobic residues in the substrate and a novel Phactr1-PP1 composite surface, which comprises a hydrophobic pocket and associated amphipathic cavity with a surrounding basic rim. These interactions allow Phactr1/PP1 to recognise its substrates 100-fold more efficiently compared with PP1 alone or the spinophilin/PP1 complex, in which the hydrophobic groove is remodelled differently.

## Results

### Crystallisation of Phactr1/PP1 complexes

Previous studies have shown that Phactr1 C-terminal sequences are necessary and sufficient for interaction with PP1 (*Allen et al., 2004*; *Wiezlak et al., 2012*): they contain an RVxF-like sequence, LIRF, and can functionally substitute for the related PP1-binding domain of yeast Bni4 (*Larson et al., 2008*; *Figure 1A*). We synthesised peptides corresponding to Phactr1(517-580) from all four Phactr family members, and measured their affinity for recombinant PP1α(7-300) using Bio-layer interferometry (BLI). Phactr1(517-580) bound with an affinity of 10.4 nM, comparable to Phactr3, while the other Phactrs bound more weakly (*Figure 1B and C*, *Figure 1—figure supplement 1A*). This $K_d$ is similar to PIPs such as spinophilin (8.7 nM) and PNUTS (9.3 nM) (*Choy et al., 2014*; *Ragusa et al., 2010*), and somewhat stronger than PPP1R15A and NIPP1 (*Choy et al., 2015*; *O'Connell et al., 2012*).

We determined the structures of purified Phactr1(507-580)/PP1α(7-300) (1.9 Å) and Phactr1(516-580)/PP1α(7-300) (1.94 Å), which crystallised at pH 8.5 and pH 5.25, respectively, and exhibit differing conformations of Phactr1 residues C-terminal to residue 567 (*Figure 1D*, *Figure 1—figure supplement 1B*). In both structures, the PP1 catalytic site contains two presumed manganese ions and a phosphate anion, as previously reported (*Figure 1—figure supplement 1C*; *Egloff et al., 1995*), and its conformation is virtually identical to that seen in other PP1 complexes, such as spinophilin/PP1 (RMSD 0.23 Å over 255 Cα atoms) (*Choy et al., 2014*; *Ragusa et al., 2010*). The Phactr1 C-terminal sequences are well resolved in the pH 8.5 structure, which is also adopted by Phactr1(516-580)/PP1α(7-300) when substrates occupy the PP1 active site, and which is supported by BLI data (discussed below). It is therefore likely be representative of the structure at physiological pH. In contrast, at pH 5.25 Phactr1 C-terminal sequences adopted a distinct and poorly resolved conformation, perhaps induced by protonation of the three C-terminal histidines (*Figure 1—figure supplement 1B*). In the discussion that follows, we focus on the pH8.5 structure.

### Phactr1 binds PP1 through an extended RVxF-φφ-R-W string

Phactr1 residues 516–542 wrap around PP1, occluding its C-terminal groove, and covering 2260 Å$^2$ of solvent-accessible surface (*Figure 1D*, *Figure 1—figure supplement 1D*), contacting PP1 in a strikingly similar way to spinophilin, PNUTS and PPP1R15B (*Chen et al., 2015*; *Choy et al., 2014*; *Ragusa et al., 2010*). These contacts include a non-canonical RVxF motif (*Egloff et al., 1997*; *Hendrickx et al., 2009*), a φφ motif (*O'Connell et al., 2012*), an Arg motif (*Choy et al., 2014*), and a previously unrecognised Trp motif (*Figure 1E–H*; *Figure 1—figure supplement 2A*). The RVxF sequence LIRF(519-522) is critical for Phactr1/PP1 complex formation (*Figure 1E*, *Figure 1—figure supplement 2A*). Its deletion decreased binding affinity ~10$^4$ fold, while the I520A and F522A mutations reduced it 4-fold and ~650 fold respectively (*Figure 1C*). The RVxF residue L519 also makes contacts with G-actin in the trivalent G-actin•Phactr1 RPEL domain complex (*Mouilleron et al., 2012*), explaining why PP1 and G-actin binding to Phactr proteins is mutually exclusive (*Figure 1—figure supplement 2B*; *Wiezlak et al., 2012*).

The other contacts also contribute to PP1-binding affinity. The φφ motif, EVAD(527-530), adds a β-strand to PP1 β-strand β14, extending one of PP1's two central β-sheets (*Figure 1F*). The Phactr1 Arg motif contacts the PP1 C-terminal groove, with R536 forming a bidentate salt bridge with PP1 D71, and a hydrogen bond with N271. This is stabilised by an intrachain cation-π interaction with Phactr1 Y534, which also hydrogen bonds with PP1 D71, and makes hydrophobic contacts with P24 and Y70 (*Figure 1G*, *Figure 1—figure supplement 2A*). Substitution of R536 by proline, as in the mouse Phactr4 'humdy' mutation (*Kim et al., 2007*), reduced Phactr1-binding affinity >300 fold, while Y534A reduced it > 10 fold (*Figure 1C*). The Trp motif, W542, which is constrained by a salt bridge between R544 and D539, makes hydrophobic contacts with PP1 I133 and Y13 (*Figure 1H*), similar to those in the PPP1R15B and spinophilin PP1 complexes (*Chen et al., 2015*; *Ragusa et al.,*

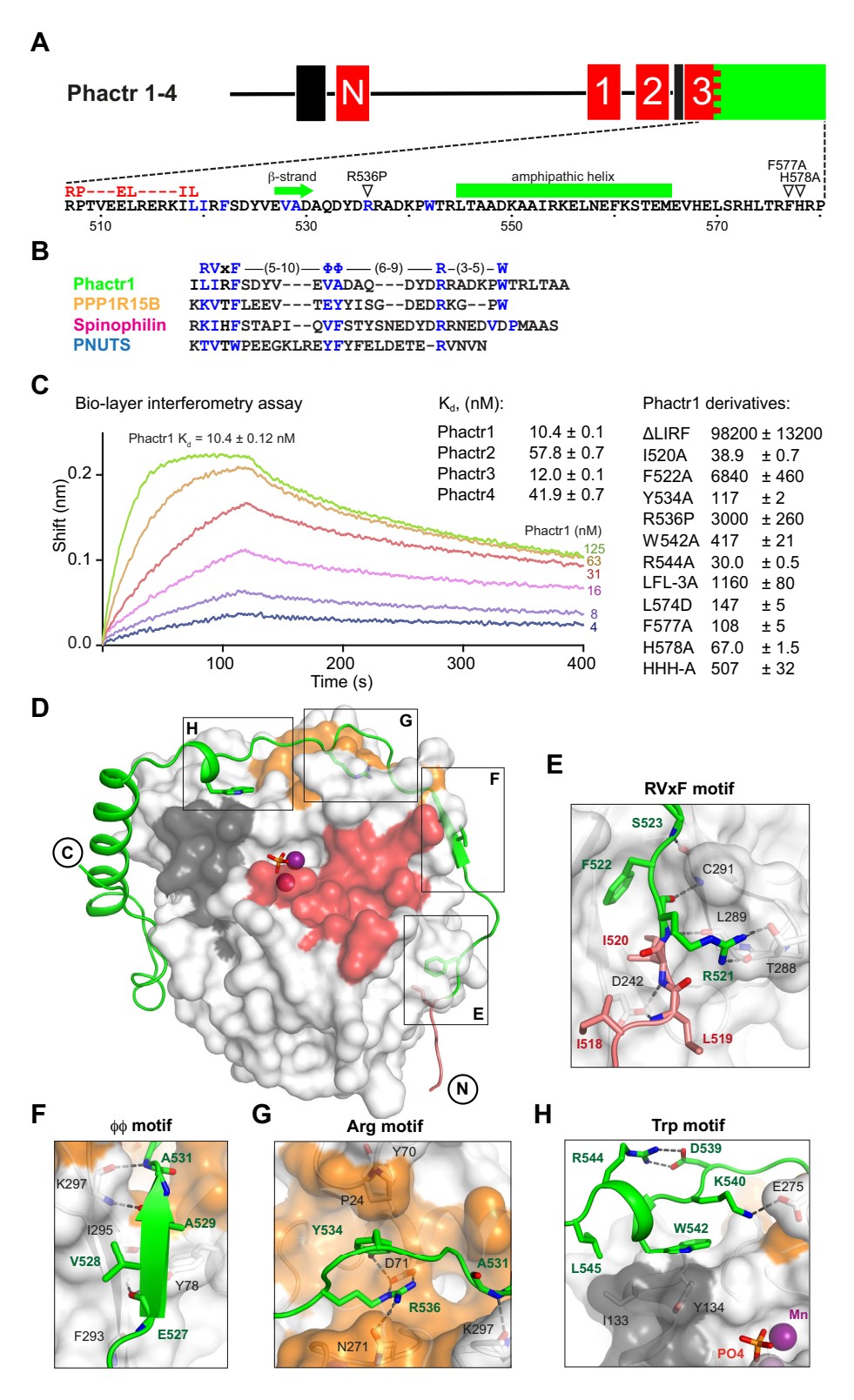

**Figure 1.** Phactr1 binds PP1 using an extended RVxF-ϕϕ-R motif. (**A**) Domain structure of Phactr family proteins. RPEL motifs, red; PP1-binding domain, green; nuclear localisation sequences, black. Below, Phactr1 C-terminal sequence, indicating RVxF-ϕϕ-R-W string, RPEL consensus, secondary structure elements, and mutations known to impair Phactr/PP1 interaction (*Allen et al., 2004*; *Kim et al., 2007*). (**B**) Structure-based alignment of Phactr1 PP1-binding sequences with other PP1 cofactors. (**C**) Bio-layer interferometry assay of PP1α(7-300) binding to Phactr1(517-580), its derivatives, and analogous

*Figure 1 continued on next page*

*Figure 1 continued*

sequences from Phactr2-4 (data are means ± SD, n = 3). (D) Structure of the Phactr1(516-580)/PP1α(7-300) complex. Phactr1, green ribbon, with RPEL motif highlighted in red, RVxF-ϕϕ-R-W sidechains as sticks; PP1, white surface, with acidic groove in red, hydrophobic groove in dark grey and C-terminal groove in gold; manganese ions, purple spheres; phosphate, orange sticks. (E–H) Detail of the individual interactions, with important residues highlighted (Phactr1, green bold; PP1, black). For comparison with other PIPs, see *Figure 1—figure supplement 2A*.

The online version of this article includes the following figure supplement(s) for figure 1:

**Figure supplement 1.** Crystallisation of Phactr1/PP1 complexes.

**Figure supplement 2.** Phactr1 binds PP1 using an extended RVxF-ϕϕ-R string.

---

*2010*; *Figure 1—figure supplement 2A*). Although W542A reduced PP1-binding affinity ~40-fold, R544A reduced it ~ threefold (*Figure 1C*).

## Phactr1 binding creates a novel composite surface adjacent to PP1 catalytic site

The Phactr1 sequences C-terminal to the RVxF-ϕϕ-R-W string form a novel structure specific to the Phactr1/PP1 complex. Residues 545–565 include a five-turn amphipathic α-helix that abuts the PP1 hydrophobic groove: Phactr1 residues L545, I553, L557, F560 make hydrophobic contacts with PP1 I133, R132, W149, and K150, while E556 and E564 make salt bridges with PP1 R132 and K150, respectively, and K561 hydrogen bonds with PP1 S129 and D194 (*Figure 2A*). Phactr1 then turns, contacting the PP1 α7-α8 loop: Phactr1 H568 hydrogen bonds to PP1 P192 and Phactr1 S571, making hydrophobic contacts with PP1 M190, while Phactr1 L570 and L574 make hydrophobic contacts with PP1 M190 and I189/P196/L201 respectively. The Phactr1 C-terminus then folds back onto the amphipathic helix, stabilised by multiple intrachain interactions. These include hydrophobic contacts between Phactr1 T575, F577 and K561; hydrogen bonds between the R576 carbonyl and K561, between the H578 amide and N558, and between the R579 carbonyl and R554, and a salt bridge between the P580 carboxylate and the R554 guanidinium. The Phactr1 C-terminus also contacts PP1 through hydrogen bonds between R576 and PP1 D194, and H578 and PP1 S129 (*Figure 2A*).

Mutagenesis experiments confirmed the importance of these interactions for Phactr1-PP1 binding (*Figure 1C*). A triple alanine substitution of Phactr1 L557, F560 and L574 (LFL-3A) resulted in a ~ 900 fold drop in binding affinity, while the acidic substitution L574D reduced binding affinity by 17-fold. Mutations F577A and H578A reduced binding activity by 16-fold and 7-fold, respectively, corroborating previous co-immunoprecipitation experiments (*Allen et al., 2004*; *Sagara et al., 2009*), and mutation of all three C-terminal histidines (HHH-A) reduced affinity >50 fold (*Figure 1C*).

The docking of Phactr1 C-terminal sequences across the PP1 hydrophobic groove substantially modifies its topography (*Figure 2B*). The positioning of the Phactr1 amphipathic helix creates a deep hydrophobic pocket comprising Phactr1 W542, L545, K550, I553, R554, L557 and H578, and PP1 I130 I133 and Y134. Adjacent to the pocket, a narrow amphipathic cavity is formed by the positioning of Phactr1 K561 and R576 across the PP1 hydrophobic groove: its polar side comprises Phactr1 K561 and R576, and PP1 D194 and S129, while its hydrophobic side is formed from PP1 residues C127, I130, V195, W206 and V223. Three water molecules and a glycerol are resolved within the cavity, whose hydrophobic side forms part of the binding site for the PP1 inhibitors tautomycetin and tautomycin (*Choy et al., 2017*). The new composite surface is crowned by a basic rim, formed from Phactr1 residues K550, R554, R576, H578 and R579 (*Figure 2B*), which radically alters the surface charge distribution (*Figure 2C*). Thus, Phactr1 binding profoundly transforms the surface of PP1 adjacent to its catalytic site. This transformation is distinct from that seen in the spinophilin/PP1 complex (*Ragusa et al., 2010*), which modifies the hydrophobic groove in a different way and leaves the PP1 surface electrostatics unchanged (*Figure 2—figure supplement 1*).

## Identification of potential Phactr1 substrates

We previously showed that expression of an activated Phactr1 derivative that constitutively forms the Phactr1/PP1 complex, Phactr1[XXX], induces F-actin rearrangements in NIH3T3 fibroblasts, provided it can bind PP1 (*Wiezlak et al., 2012*). Indeed, overexpression of the Phactr1 PP1-binding domain alone can also induce such changes, suggesting that the Phactr1 C-terminal sequences are sufficient to allow recognition of at least some substrates (*Figure 3—figure supplement 1A*; see

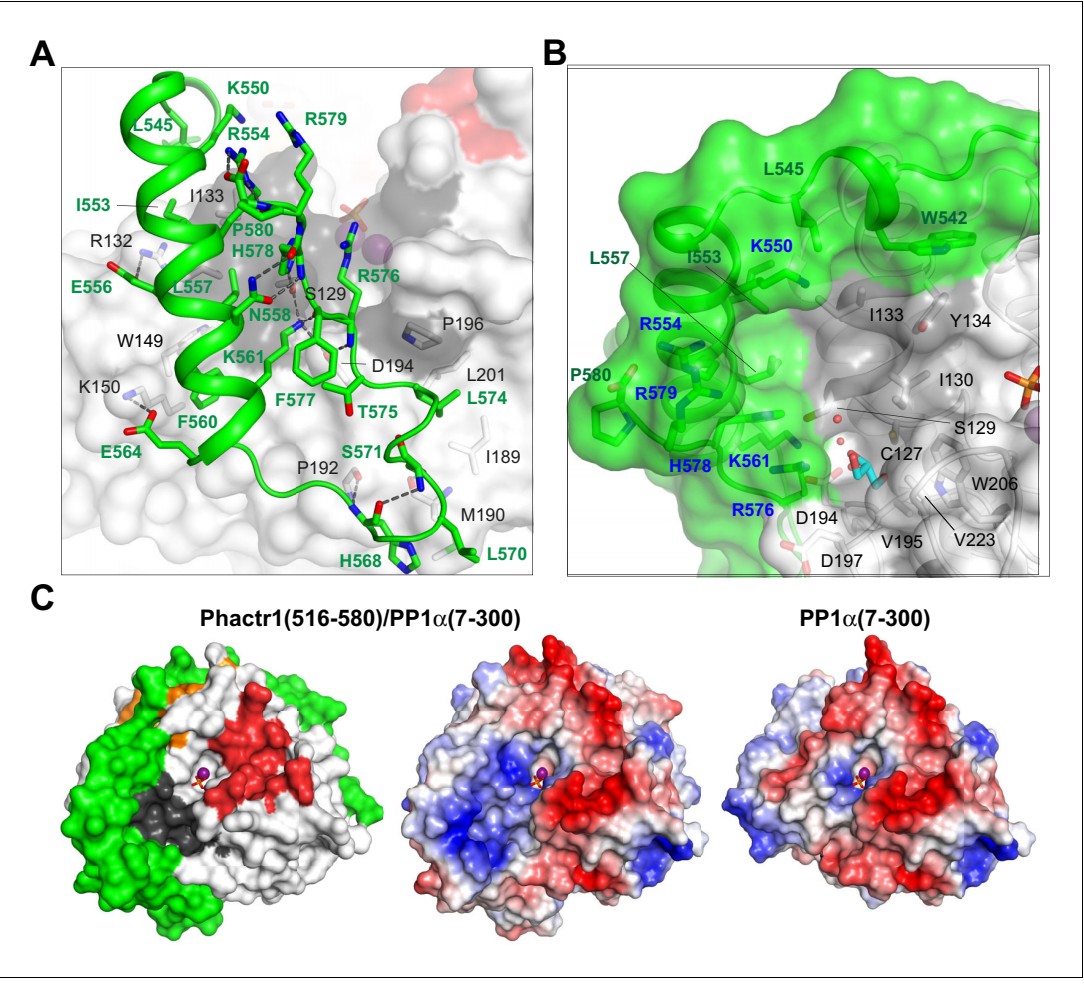

**Figure 2.** Phactr1/PP1 interaction remodels the PP1 hydrophobic groove. (**A**) Molecular interactions by the Phactr1 C-terminal sequences (green; residues involved in intrachain interactions or contacting PP1 are shown as sticks). (**B**) The novel composite surface formed by Phactr1/PP1 interaction, showing the deep hydrophobic pocket, adjacent narrow amphipathic cavity with associated waters (red spheres) and glycerol (cyan sticks), and residues constituting the Phactr1-derived basic rim (blue). (**C**) PP1 surface electrostatics are transformed in the Phactr1/PP1 complex. Left, surface representation of the Phactr1/PP1 complex; centre, electrostatic surface potential representation of Phactr1/PP1 complex. Right, electrostatic surface potential representation of PP1 (positive, blue; negative, red). The online version of this article includes the following figure supplement(s) for figure 2:

**Figure supplement 1.** Spinophilin remodels the PP1 hydrophobic groove in a different way to Phactr1.

---

Discussion). These observations suggest that the Phactr1/PP1 complex might dephosphorylate target proteins involved in cytoskeletal dynamics. To identify potential Phactr1/PP1 substrates, we used differential SILAC phosphoproteomics in NIH3T3 cells expressing Phactr1[XXX]. Over 3000 phosphorylation sites were quantified, and each assigned a dephosphorylation score comparing their phosphorylation with that observed in cells inducibly expressing Phactr1[XXX]ΔC, which lacks the PP1 binding sequences, or vector alone (*Figure 3A*; *Figure 3—figure supplement 1B*, *Supplementary file 1*, Table A).

Annotation enrichment analysis of the whole dataset (*Cox and Mann, 2012*) identified two Gene Ontology Biological Process categories that exhibited a significantly higher mean dephosphorylation score upon expression of Phactr1[XXX]: 'regulation of actin filament-based process' and 'regulation of actin cytoskeleton organisation' (*Figure 3B*, *Supplementary file 1*, Table B). In keeping with this, proteins with a high dephosphorylation score included many cytoskeletal components and regulators (*Figure 3C*, *Supplementary file 1*, Table A). Proline-directed sites predominated in the dataset

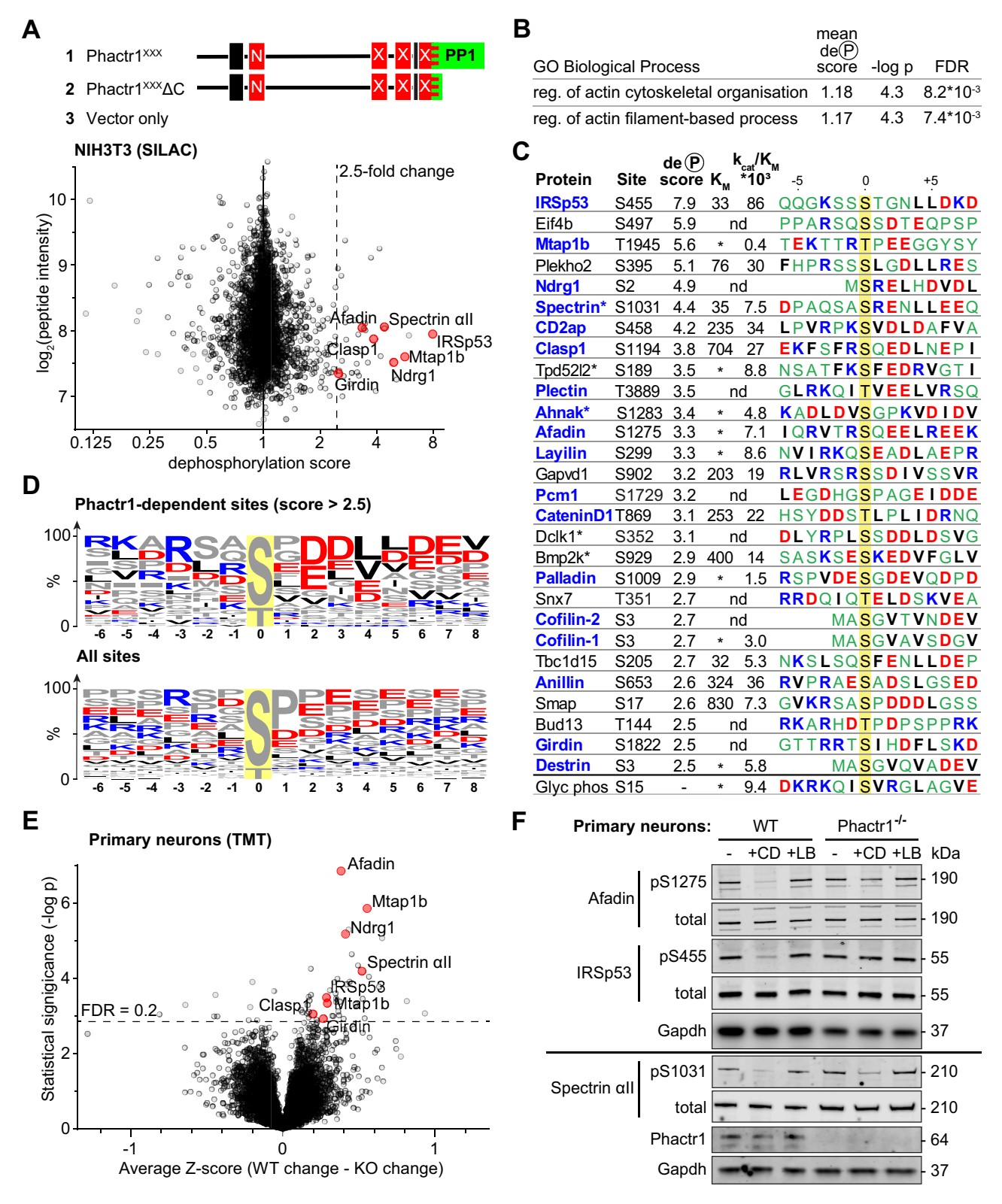

**Figure 3.** Identification of Phactr1/PP1 substrates. (A–D), NIH3T3 cells, (E, F) neuronal cells. (A) SILAC phosphoproteomics in NIH3T3 cells. NIH3T3 cell lines conditionally expressed Phactr1[XXX] (constitutively binds PP1, not G-actin), Phactr1[XXX]ΔC (binds neither PP1 nor G-actin), or vector alone (*Wiezlak et al., 2012*). Forward and reverse SILAC phosphoproteomics were used to generate a dephosphorylation score, quantified below (red highlights, hits also detected as Phactr1-dependent in neurons). (B) Annotation enrichment analysis of the entire SILAC phosphoproteomics dataset for

*Figure 3 continued on next page*

*Figure 3 continued*

GO Biological Process terms, showing terms with dephosphorylation score >1, FDR < 0.02. (C) Candidate Phactr1/PP1 substrates ranked by dephosphorylation score (blue, cytoskeletal structural or regulatory proteins; asterisks, proteins with multiple dephosphorylation sites). Michaelis constant ($K_M$, μM), specificity constant ($k_{cat}/K_M$, $μM*min^{-1}*U^{-1}$) and sequence context (blue, basic; red, acidic; black, hydrophobic) are shown. (*), $K_M$ could not be reliably determined; (nd), not done. (D) Amino acid frequency among phosphorylation sites with dephosphorylation score >2.5 (top) compared with all phosphorylation sites (bottom). (E) Phactr1-dependent protein dephosphorylation in cortical and hippocampal neurons treated with cytochalasin D (CD). Differential Z-score, the difference between the phosphorylation change observed in Phactr1-wildtype and Phactr1-null neurons, plotted versus statistical significance. Red highlights, peptides also observed in the NIH3T3 SILAC phosphoproteomics. (F) Validation of TMT phosphoproteomics data in primary cortical neurons treated for 30' with CD or latrunculin B (LB). For quantitation, see *Figure 3—figure supplement 2D*. See also *Supplementary file 3*.

The online version of this article includes the following figure supplement(s) for figure 3:

**Figure supplement 1.** Substrates of the Phactr1/PP1 complex in NIH3T3 fibroblasts.
**Figure supplement 2.** Neuronal substrates of the Phactr1/PP1 complex.

as a whole, but those sites with dephosphorylation scores > 2.5 were enriched in acidic residues at positions +2 to +7 relative to the phosphorylation site, with small hydrophobic residues enriched at positions +4 and +5, and basic residues N-terminal to it (*Figure 3D*). Since this sequence bias was not observed at all sites, we tested directly whether sites were Phactr1/PP1 substrates using an in vitro peptide dephosphorylation assay. Synthetic phosphopeptides containing the candidate sites exhibited a range of $K_M$ and $k_{cat}/K_M$ values: for example, IRSp53 pS455 exhibited a low $K_M$ and high $k_{cat}/K_M$; spectrin αII pS1031 had a similar $K_M$ but lower $k_{cat}/K_M$; and afadin pS1275 was as reactive as spectrin αII pS1031, but with much poorer $K_M$ (*Figure 3C*; *Figure 3—figure supplement 1C*; *Supplementary file 2*). Interestingly, the four substrates with the lowest $K_M$ - IRSp53 pS455, Plekho2 pS395, spectrin αII pS1031, and Tbc1d15 pS205 - all contained a leucine doublet at the +4 and +5 positions (see Discussion).

## IRSp53, afadin, and spectrin αII are Phactr1/PP1 substrates

We generated phosphospecific antisera against IRSp53 pS455, spectrin αII pS1031, and afadin pS1275. Immunoblot analysis showed that in NIH3T3 cells, phosphorylation of IRSp53 S455 and afadin S1275 was substantially decreased upon expression of Phactr1[XXX], and by serum stimulation, which activates rho-actin signalling and Phactr-family/PP1 complex formation (*Miralles et al., 2003*; *Vartiainen et al., 2007*; *Wiezlak et al., 2012*; *Figure 3—figure supplement 1D*; *Supplementary file 3*). Moreover, treatment of cells with cytochalasin D (CD), which binds G-actin and disrupts its interaction with RPEL proteins (*Vartiainen et al., 2007*; *Wiezlak et al., 2012*) significantly decreased phosphorylation of IRSp53 S455 and afadin S1275 (*Figure 3—figure supplement 1E*; *Supplementary file 3*). In contrast, treatment of cells with the latrunculin B (LB), which increases G-actin concentration through blockade of F-actin assembly, but whose binding to G-actin does not affect RPEL-actin interaction, had the opposite effect (*Figure 3—figure supplement 1E*; *Supplementary file 3*). These data suggest that endogenous RPEL protein(s), presumably Phactr-family members, control IRSp53 S455 and afadin S1275 phosphorylation in NIH3T3 cells.

To explore directly the specific involvement of Phactr1 in protein dephosphorylation we turned to neurons, which express Phactr1 at high level (*Allen et al., 2004*). Phactr1 mutations are associated with morphological and functional developmental defects in cortical neurons (*Hamada et al., 2018*), and expression of Phactr1[XXX] induced morphological defects upon expression in cultured hippocampal neurons (*Figure 3—figure supplement 2A*). To assess whether Phactr1 controls phosphorylation in neurons, we analysed hippocampal and cortical neurons from wildtype and Phactr1-null animals. Phactr1-null mice are viable; they do not show any obvious developmental abnormalities, and expression of the other Phactr proteins is apparently unaffected (*Figure 3—figure supplement 2B*). Neurons were treated with LB or CD to inhibit or activate RPEL proteins, and phosphorylation profiles analysed using TMT phosphoproteomics. Among ~9000 phosphorylation sites quantified, we found 44 sites on 37 proteins that differed significantly in their response to these stimuli in Phactr1-null cells (*Figure 3E*, *Supplementary file 1*, Table E). The sequence context of these sites was similar to that of those seen in NIH3T3 cells (*Figure 3—figure supplement 2C*), and seven, including IRSp53 pS455, spectrin αII pS1031, and afadin pS1275, were observed in both cell types (*Figure 3F*, *Figure 3—figure supplement 2D*; *Supplementary file 3*). In sum, these results identify multiple

Phactr1 substrates in NIH3T3 cells and neurons, many of which are cytoskeletal components or regulators.

## Crystallisation of Phactr1/PP1-substrate complexes

To understand the molecular basis for substrate recognition by Phactr1/PP1, we sought to determine the structures of Phactr1/PP1-substrate complexes. We were unable to co-crystallise Phactr1/PP1 with unphosphorylated or glutamate-phosphomimetic peptides encompassing IRSp53 S455, afadin S1275 or spectrin αII S0131, so we used a PP1-substrate fusion strategy similar to that used for PP5 (*Oberoi et al., 2016*). Fusion proteins comprising PP1(7–304) joined via a $(SG)_9$ linker to unphosphorylated substrate sequences were coexpressed with Phactr1(516-580) for structural analysis. This approach allowed determination of the structures of the IRSp53(449-465) and spectrin αII (1025–1039) complexes at 1.09 Å and 1.30 Å resolution, respectively (*Figure 4A–C*; *Table 1*; *Figure 4—figure supplement 1*; referred to hereafter as the IRSp53 and spectrin complexes), but was unsuccessful with Tbc1d15, Plekho2, afadin, and cofilin.

The Phactr1/PP1-IRSp53 and Phactr1/PP1-spectrin complexes crystallised in the same spacegroup. Each asymmetric unit contains two complexes (RMSD 0.22 Å and 0.17 Å over 288 Cα, respectively), in which substrate-Phactr1/PP1 interactions are mostly conserved, albeit with some minor differences (*Figure 4—figure supplement 1A and B*). The substrate sequences, whose N-termini are largely unresolved, make extensive contacts across the PP1 catalytic site, then extend in a sinuous trajectory across the composite Phactr1/PP1 surface, making numerous hydrophobic and ionic contacts (*Figure 4C*). Like the Phactr1/PP1 holoenzyme, the complexes contain a presumed phosphate anion at the catalytic site, and thus appear to represent putative enzyme/product complexes (see below). The structure of Phactr1/PP1 in both complexes is identical to that of the isolated Phactr1/PP1 complex (RMSD 0.31 Å over 313 Cα and 0.32 Å over 307 Cα, respectively).

## Phactr1/PP1-substrate interactions are opposite in polarity to those of PP5

IRSp53 and spectrin make virtually identical contacts with the PP1 catalytic cleft, predominantly via their mainchains (*Figure 4C*; *Figure 4—figure supplement 1C*). In IRSp53 complex 2, K452$^{IRSp53}$ (−3 relative to the phosphorylation site) makes a salt bridge with PP1 D220 carboxyl, in the acidic groove, while water-bridged hydrogen bonds link the mainchain carbonyl and the sidechain hydroxyl of S453$^{IRSp53}$/S1029$^{spectrin}$(−2) to PP1 Y272 and the phosphate, and to V250, respectively. The dephosphorylated S455$^{IRSp53}$/S1031$^{spectrin}$(0) hydroxyl interacts with PP1 R96, H125 and the phosphate, its mainchain amide and carbonyl contacting the phosphate and PP1 R221, respectively (*Figure 4A–C*, *Figure 4—figure supplement 1A and B*). The phosphate is inverted compared with the holoenzyme complex, losing its contact with PP1 R96 and H125, but making contact with the S455$^{IRSp53}$/S1031$^{spectrin}$ hydroxyl (*Figure 4D*; *Figure 4—figure supplement 1A–C*). C-terminal to the dephosphorylated serine, T456$^{IRSp53}$/R1032$^{spectrin}$(+1) hydrogen bonds with PP1 Y134 via its mainchain amide, with R1032$^{spectrin}$ making two additional hydrogen bonds, with PP1 Y134 via its carbonyl, and the D220 carbonyl via its sidechain.

The majority of the PP1 catalytic cleft residues that contact IRSp53 and spectrin are conserved among PPP family members, including PP5, which has been crystallised in complex with a phosphomimetic derivative of its substrate, Cdc37(S13E) (*Oberoi et al., 2016*). Strikingly, in that complex, the PP5 residues corresponding to PP1 R96, H125, Y134, R221, and Y272 make contacts with Cdc37 (S13E) analogous to those seen in the Phactr1/PP1-substrate complexes, despite the fact that Cdc37 (S13E) docks in the PP5 catalytic cleft in the opposite orientation to IRSp53 and spectrin (*Figure 4—figure supplement 2*; see Discussion).

## Catalytic mechanism

It is generally accepted that the phosphate seen in many PPP family protein structures binds the active site in a similar way to the substrate phosphate (*Egloff et al., 1995*; *Griffith et al., 1995*; *Mueller et al., 1993*; *Swingle et al., 2004*), and we therefore assume that the phosphate in our Phactr1/PP1 holoenzyme structure is positioned similarly to the phosphorylated S455$^{IRSp53}$/S1031$^{spectrin}$ of the bound substrate (*Figure 4D*). Consistent with this idea, in the PP5/Cdc37(S13E) complex, the phosphomimetic glutamate sidechain carbonyl is virtually superposable on the

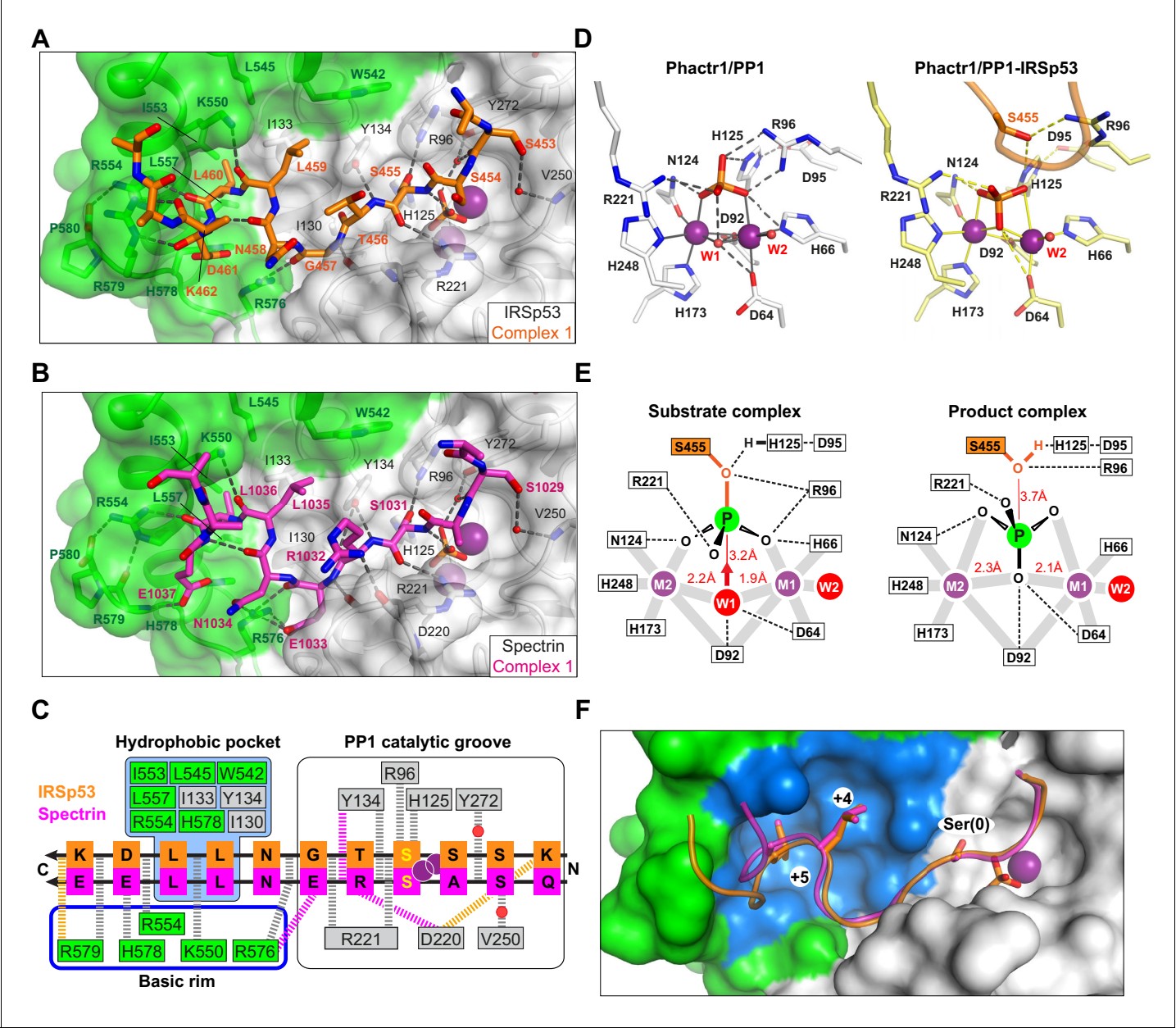

**Figure 4.** Substrate interactions with the Phactr1/PP1 holoenzyme. (**A, B**) Structures of (**A**) the Phactr1/PP1-IRSp53(449-465) and (**B**) the Phactr1/PP1-spectrin(1025–1039) complexes, displayed as in *Figure 1*, with IRSp53 and spectrin displayed in orange and magenta sticks, respectively. (**C**) Summary of substrate interactions. Hydrogen bonds are shown as thick dashed lines: grey for both substrates; colour, for specific substrate. Composite hydrophobic surface residues are highlighted in blue (see **F**). (**D**) Inversion of the recruited phosphate. Phosphate and metal ion contacts in the Phactr1/PP1 and in Phactr1/PP1-IRSp53 structures are shown. Metal coordination bonds, solid continuous lines; hydrogen bonds, dashed lines; W1 and W2, water molecules. (**E**) Potential catalytic mechanism. Left, a hypothetical substrate complex, based on the Phactr1/PP1 complex, assuming that its phosphate corresponds to that of IRSp53 pS455. Right, the observed Phactr1/PP1-IRSp53 product complex. W1 and W2, water molecules; grey bars, metal coordination bonds; dashes, hydrogen bonds. Proposed nucleophilic attack by activated W1 results in phosphate inversion. (**F**) Docking of the SxxxLL motif (sticks) with the Phactr1/PP1 hydrophobic pocket. Phactr1/PP1 in surface representation, with the composite hydrophobic surface in light blue, and other Phactr1 and PP1 surfaces in green and white, respectively.

The online version of this article includes the following figure supplement(s) for figure 4:

**Figure supplement 1.** Substrate interactions in the Phactr1/PP1 complex.
**Figure supplement 2.** Opposite polarity of substrate binding to PP5 and Phactr1/PP1.
**Figure supplement 3.** Comparison of the PP5-Cdc37(S13E), Phactr1/PP1 and Phactr1/PP1-IRSp53 structures gives insight into catalytic mechanism.

**Table 1.** Crystallographic data and refinement statistics.

| | Phactr1/PP1 (pH 8.5) | Phactr1/PP1 (pH 5.25) | Phactr1/PP1-IRSp53 (pH 8.5) | Phactr1/PP1-spectrin-αII (pH 8.5) | Phactr1/PP1-IRSp53-S455E (pH 8.5) | PP1-Phactr1(526-580) (pH 8.5) |
|---|---|---|---|---|---|---|
| PDB ID | 6ZEE | 6ZEF | 6ZEG | 6ZEH | 6ZEI | 6ZEJ |
| Resolution range | 238.79–1.90 (1.93–1.90) | 56.05–1.94 (2.01–1.94) | 35.12–1.09 (1.13–1.09) | 48.47–1.30 (1.35–1.30) | 69.37–1.39 (1.44–1.39) | 59.48–1.78 (1.84–1.78) |
| Space group | P6₅ | P 1 | P 2₁ | P 2₁ | P 2₁ | P 6₅ |
| Unit cell a, b, c | 137.7 137.7 238.8 | 47.5 57.5 89.6 | 48.6 122.3 69.0 | 48.5 122.3 69.3 | 48.7 122.3 69.4 | 137.37 137.37 238.02 |
| α, β, γ | 90 90 120 | 78.0 74.6 81.6 | 90 92.2 90 | 90 92.1 90 | 90 92.2 90 | 90 90 120 |
| Total reflections | 2 420 272 (99 345) | 522 449 (32 609) | 664 254 (62 356) | 1 278 289 (123 745) | 1 058 251 (79 018) | 5 001 498 (505 725) |
| Unique reflections | 200 820 (9 938) | 65 136 (6 061) | 332 349 (31 383) | 196 809 (19 297) | 162 314 (15 982) | 241 814 (24 087) |
| Multiplicity | 12.1 (10.0) | 8.0 (5.4) | 2.0 (2.0) | 6.5 (6.4) | 6.5 (4.9) | 20.7 (21.0) |
| Completeness (%) | 100 (100) | 98.5 (92.0) | 99.3 (93.7) | 99.1 (96.8) | 99.6 (98.3) | 99.7 (99.6) |
| Mean I/sigma(I) | 7.5 (1.1) | 9.7 (2.7) | 15.4 (1.8) | 5.9 (1.1) | 12.5 (1.1) | 12.9 (1.1) |
| Wilson B-factor | 33.1 | 30 | 10.8 | 11.1 | 16.4 | 26.4 |
| R-merge | 0.14 (2.1) | 0.13 (0.57) | 0.01 (0.44) | 0.13 (1.78) | 0.06 (1.05) | 0.179 (3.19) |
| R-meas | 0.16 (2.3) | 0.14 (0.64) | 0.02 (0.62) | 0.15 (1.94) | 0.07 (1.18) | 0.18 (3.26) |
| R-pim | 0.04 (0.75) | 0.04 (0.26) | 0.01 (0.44) | 0.05 (0.75) | 0.02 (0.52) | 0.04 (0.70) |
| CC1/2 | 0.99 (0.70) | 0.99 (0.88) | 1.0 (0.65) | 0.99 (0.47) | 1.0 (0.54) | 1.0 (0.54) |
| Reflections used in refinement | 199 843 (6 288) | 65 085 (6 061) | 332 111 (31 289) | 195 791 (19 135) | 162 184 (15 972) | 241 401 (24 054) |
| Reflections used for R-free | 10 138 (347) | 3 132 (295) | 16 696 (1 543) | 9 611 (927) | 1 999 (197) | 12 175 (1 276) |
| R-work | 0.23 (0.34) | 0.18 (0.22) | 0.11 (0.23) | 0.13 (0.23) | 0.12 (0.24) | 0.24 (0.36) |
| R-free | 0.26 (0.37) | 0.21 (0.24) | 0.14 (0.24) | 0.16 (0.26) | 0.15 (0.27) | 0.27 (0.40) |
| Number of non-hydrogen atoms | 18 201 | 5 962 | 7 225 | 6 778 | 6 822 | 17 414 |
| Macromolecules | 17428 | 5 683 | 6 339 | 6 104 | 6 099 | 16 682 |
| Ligands | 33 | 37 | 42 | 19 | 32 | 58 |
| Solvent | 740 | 242 | 844 | 655 | 691 | 674 |
| Protein residues | 2160 | 715 | 754 | 742 | 742 | 2 094 |
| RMS(bonds) | 0.011 | 0.005 | 0.01 | 0.01 | 0.011 | 0.011 |
| RMS(angles) | 0.85 | 0.84 | 1.17 | 1.15 | 1.29 | 0.84 |
| Ramachandran favoured (%) | 96 | 95.62 | 97.15 | 96.99 | 96.99 | 95.85 |
| Ramachandran allowed (%) | 4 | 4.38 | 2.85 | 3.01 | 2.88 | 4.15 |
| Ramachandran outliers (%) | 0 | 0 | 0 | 0 | 0.14 | 0 |
| Clashscore | 2.51 | 2.65 | 4.7 | 3.05 | 3.87 | 2.7 |
| Average B-factor | 44.4 | 41 | 16.2 | 19 | 21.7 | 41 |
| Macromolecules | 44.9 | 40.8 | 14.6 | 17.6 | 20.1 | 41.1 |
| Ligands | 48.7 | 46.7 | 23.6 | 14.9 | 23.7 | 44.7 |
| Solvent | 41.6 | 43.1 | 28 | 32.8 | 35.4 | 37.1 |

phosphate in the Phactr1/PP1 complex (*Figure 4—figure supplement 3*). The Phactr1/PP1 complex also contains a bound water, W1, presumably activated by the metal ions and PP1 D64 and D92 (*Figure 4D*). W1 is oriented appropriately for in-line nucleophilic attack on the substrate phosphate: this would be facilitated by protonation of the phosphoserine oxygen by the PP1 catalytic dyad H125-D95, and would result in inversion of the phosphate (*Figure 4E*, *Figure 4—figure supplement*

3). The structures of our Phactr1/PP1-IRSp53/spectrin complexes are consistent with their representing the resulting enzyme-product complexes, stablised through the tethering of the substrate sequences to the holoenzyme. The structures thus provide direct evidence for the in-line nucleophilic hydrolysis mechanism for PPP family phosphatases, as outlined by the Barford and Ciszak groups (*Egloff et al., 1995*; *Swingle et al., 2004*).

## Substrates make extensive contacts with the novel Phactr1/PP1 surface

C-terminal to their interaction with the catalytic cleft, both IRSp53 and spectrin follow similar trajectories from residues +2 to +7 (*Figure 4A–C*, *Figure 4—figure supplement 1C*). The most striking feature of the complexes is the multiplicity of contacts made between the substrate and the novel composite Phactr1/PP1 surface created by extension of the PP1 hydrophobic groove, most of which are conserved between the two substrates (*Figure 4C and F*). In both structures, the residues +3 to +6 form a β-turn, allowing the hydrophobic doublet L459-L460$^{\text{IRSp53}}$/L1035-L1036$^{\text{spectrin}}$ (+4/+5) to make intimate contact with the Phactr1/PP1 hydrophobic pocket, entirely burying the +5 leucine sidechain (*Figure 4C and F*). These interactions are stabilised by hydrogen bonds between the mainchain carbonyls of substrate residues +2, +4, +5 and Phactr1 R576 K550, and R554, and between the sidechain of the substrate acidic +6 residue (D461$^{\text{IRSp53}}$/E1037$^{\text{spectrin}}$) and Phactr1 H578 (*Figure 4A–C*); a hydrogen bond between the D461$^{\text{IRSp53}}$ mainchain carbonyl and Phactr1 R579 is substituted by a hydrogen bond between the E1037$^{\text{spectrin}}$ carboxylate and Phactr1 R579 mainchain amide. At position +2 (G457$^{\text{IRSp53}}$/E1033$^{\text{spectrin}}$), the mainchain amide and carbonyl make hydrogen bonds with the mainchain carbonyls of PP1 R221 and Phactr1 R576 respectively; in addition, the E1033$^{\text{spectrin}}$ side chain spans the top of the amphipathic cavity to make an additional salt bridge with the Phactr1 R576 side chain (*Figure 4B and C*).

## Phactr1/PP1 hydrophobic pocket interactions promote catalytic efficiency

To investigate the functional significance of the Phactr1/PP1-substrate interactions seen in the structures, we tested mutated IRSp53 peptide substrates in the in vitro dephosphorylation assay (*Figure 5A*). Alanine substitutions at positions S453 (−2), S454 (−1), and T456 (+1) decreased catalytic efficiency somewhat, apparently reflecting an increased $K_M$. In contrast, alanine substitution of residues L459, L460, and D461 (+4 to +6) individually increased $K_M$, but had variable effects on catalytic efficiency, with L459A and D461A increasing it, and L460A reducing it (see Discussion). Pairwise combination of substitutions at positions +4 through +6 suppressed reactivity even more, with the LL459/460AA mutant being essentially unreactive. Alanine substitution at K462 (+7) and D463 (+8) either decreased or increased $K_M$ with corresponding effects on catalytic efficiency, consistent with a preponderance of acidic residues at positions +6 to +8 (see *Figure 3D*). Upon expression in NIH3T3 cells, IRSp53(L460A) exhibited enhanced basal phosphorylation, and was less susceptible to CD-induced dephosphorylation than wildtype IRSp53, as assessed using the phospho-S455 antibody (*Figure 5B*). These data show that substrate interactions with the Phactr1/PP1 hydrophobic pocket are important for efficient dephosphorylation both in vitro and in vivo.

The effects of substrate alanine mutations on $K_M$ suggests that they affect the affinity of Phactr1/PP1-substrate interactions. To investigate substrate binding directly, we used a peptide overlay assay in which unphosphorylated substrate peptides were immobilised on membranes, and tested for their ability to recruit recombinant GST-Phactr1(516-580)/PP1(7–300) complex from solution. Since these peptides lack phosphoserine, it is likely that their binding will predominantly be determined by interactions outside the catalytic site. Only peptides representing low $K_M$ substrates - IRSp53, spectrin αII, Tbc1d15 and Plekho2 - exhibited detectable binding under the conditions of the assay (*Figure 5—figure supplement 1*; see *Figure 3C*). To assess the contribution of individual residues to binding, we used IRSp53 and spectrin arrays in which each substrate residue was systematically changed to every other amino acid (*Figure 5C and D*). Tryptophan and cysteine substitutions at non-critical residues led to a general increase in binding affinity, and we therefore did not attempt to interpret these substitutions. Analysis of both arrays implicated residues −2 to +7 in substrate-binding affinity. Positions +4 and +5 displayed the strongest selectivity, for hydrophobic residues and leucine respectively: indeed, phenylalanine substitution of IRSp53 L459 (+4), increased binding affinity and catalytic efficiency (*Figure 5A*). In contrast, the sequence dependence at other positions

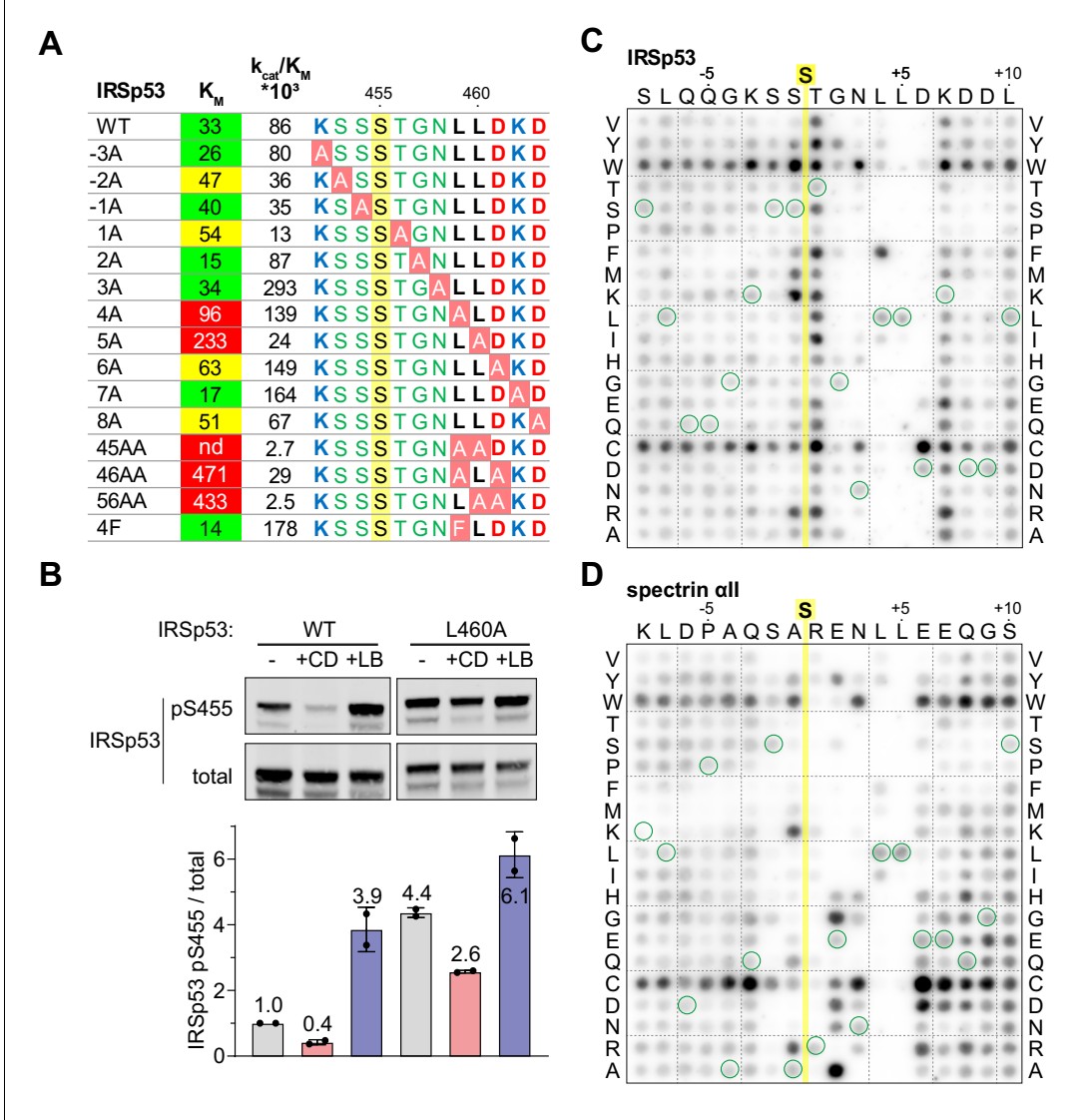

**Figure 5.** Efficient dephosphorylation involves substrate interaction with the Phactr1/PP1 composite surface. (A) Phactr1/PP1 dephosphorylation of alanine substitution derivatives of IRSp53 S455 substrate 19mer phosphopeptides. $K_M$ values are highlighted: green,<40 µM; yellow, 40–80 µM; red,>80 µM. (B) Immunoblot analysis of total IRSp53 and IRSp53 phospho-S455 levels after expression of wild-type IRSp53 or IRSp53 L460A in NIH3T3 cells with 30' CD or LB treatment as indicated. (C,D) Overlay binding affinity assay of IRSp53 (C) and spectrin αII (D). Arrays contained the variants of the wild-type sequence, in which each amino acid is systematically changed to each other amino acid as indicated vertically, with wild-type sequence circled in green. Yellow line, position of the invariant unphosphorylated target serine.

The online version of this article includes the following figure supplement(s) for figure 5:

**Figure supplement 1.** Binding of candidate Phactr1/PP1 substrates in the peptide array overlay assay.

was strongly context-dependent. In the vicinity of the target residue, IRSp53 binding was relatively unaffected by N-terminal variation, while T456 (+1) was suboptimal, with basic or hydrophobic residues being preferred; in contrast, spectrin binding was strongly selective at N-terminal positions, with a preference for basic residues at −1 and +1, and serine at −2. Similarly, IRSp53 exhibited a strong preference for acidic residues at position +6, perhaps reflecting the presence of the suboptimal neighbouring basic residue, K462 (+7), which was not the case for spectrin. Taken together with the kinetics data, these results show that interactions with the Phactr1/PP1 hydrophobic pocket are a critical determinant of Phactr1/PP1 substrate recognition.

## Variable spacing between the dephosphorylation site and hydrophobic residues

In IRSp53, the dephosphorylated residue S455 is flanked by three other potential phosphoacceptors, S453, S454, and T456. We considered the possibility that strong interactions with the Phactr1/PP1 hydrophobic pocket might also allow efficient dephosphorylation of these residues. That this might be the case was suggested by our structural analysis of a phosphomimetic fusion construct, in which the S455 is substituted by glutamate, at 1.39 Å resolution (Phactr1/PP1-IRSp53(S455E)). Strikingly, in this complex, the phospomimetic glutamate (S455E) does not replace phosphate at the catalytic site. Instead, IRSp53 T456 occupies the position corresponding to S455 in the wildtype complex, with a recruited phosphate positioned as in the wild-type complex, and the glutamate (S455E) in effect occupies position −1, pointing away from the catalytic site (*Figure 6A*). Nevertheless, the critical contacts between IRSp53(S455E) L459/L460 and the Phactr1/PP1 hydrophobic pocket are maintained, extending the intervening sequence G457-N458-L459 (*Figure 6B*). Consistent with this, both wildtype IRSp53 pT456 and IRSp53 E455/pT456 were effective substrates for Phactr1/PP1, exhibiting three- to fivefold higher $K_M$, but only twofold lower catalytic efficiency, while IRSp53 pS453 and pS454 were very poor substrates, exhibiting 10–30 fold lower catalytic efficiency (*Figure 6C*). Thus, Phactr1/PP1 substrates can tolerate either three or four residues between the target residue and the hydrophobic residue that engages the Phactr1/PP1 hydrophobic pocket (see Discussion).

## Interaction with Phactr1 confers substrate specificity on PP1

Since the composite hydrophobic surface of the Phactr1/PP1 holoenzyme plays an important role in substrate binding and catalytic efficiency, we next investigated to what extent interaction with Phactr1 confers substrate specificity on PP1. We were particularly interested to test whether the high $K_M$ and lower binding affinity of Phactr1/PP1 substrates such as afadin and cofilin might be associated with an increased promiscuity in their reactivity with different PP1 complexes. In addition to comparing Phactr1/PP1 with apo-PP1, we therefore also compared it with the spinophilin/PP1α complex (*Figure 1—figure supplement 2A*, *Figure 6—figure supplement 1A*; hereafter spinophilin/PP1) (*Ragusa et al., 2010*). As discussed above, spinophilin also interacts with PP1 through an extended RVxF-ϕϕ-R-W string, but remodels the hydrophobic groove differently, and does not change PP1 surface electrostatics (*Figure 2—figure supplement 1*; *Ragusa et al., 2010*). We also assessed the substrate specificity of a fusion protein in which the Phactr1 residues 526–580 were fused via a short SG linker to PP1 residue 304, solved at 1.78 Å resolution (*Figure 6D*). This fusion protein lacks the RVxF-PP1 interaction critical for formation of the authentic Phactr1/PP1 complex, but nevertheless generates a composite hydrophobic surface virtually identical to that seen in the Phactr1/PP1 holoenzyme (RMSD 0.25 Å over 2395 atoms; *Figure 6—figure supplement 1B*).

We tested a series of Phactr1/PP1 substrate phosphopeptides with decreasing catalytic efficiency and increasing $K_M$, and analogous phosphopeptides from glycogen phosphorylase and GluR1, substrates of the GM/PP1 and spinophilin/PP1 complexes (*Hu et al., 2007*; *Ragusa et al., 2010*). Compared with apo-PP1 and spinohilin/PP1, Phactr1/PP1 exhibited 100- to 400-fold greater catalytic efficiency against its substrates IRSp53 pS455, CD2ap pS458, and afadin pS1275 (*Figure 6E*). However, while Phactr1/PP1 dephosphorylated cofilin pS3 with a low catalytic efficiency comparable to afadin pS1275, this was only twofold enhanced compared with PP1 or spinophilin/PP1 (*Figure 6E*; see Discussion). Thus, at least some Phactr1 substrates with high $K_M$, are likely to be substates for multiple different PP1 complexes (see Discussion). Phactr1 did not enhance the activity of PP1 against the GM/PP1 substrate glycogen phosphorylase pS15, or the spinophilin/PP1 target GluR1 pS863 (*Figure 6E*, *Figure 6—figure supplement 1C*). The PP1-Phactr1 fusion protein behaved similarly to the Phactr1/PP1 complex, indicating that RVxF interactions are not essential for specificity (*Figure 6E*). We found that spinophilin did not enhance PP1 activity against the short GluR1 pS863 peptide (*Figure 6—figure supplement 1C*), although it is active against a longer GluR1 fragment (*Ragusa et al., 2010*), suggesting that substrate specificity is not directed by its modified hydrophobic groove. Taken together with the results in the preceding sections, these results show that the composite surface formed by interaction with Phactr1 is responsible for the substrate specificity of the Phactr1/PP1 holoenzyme.

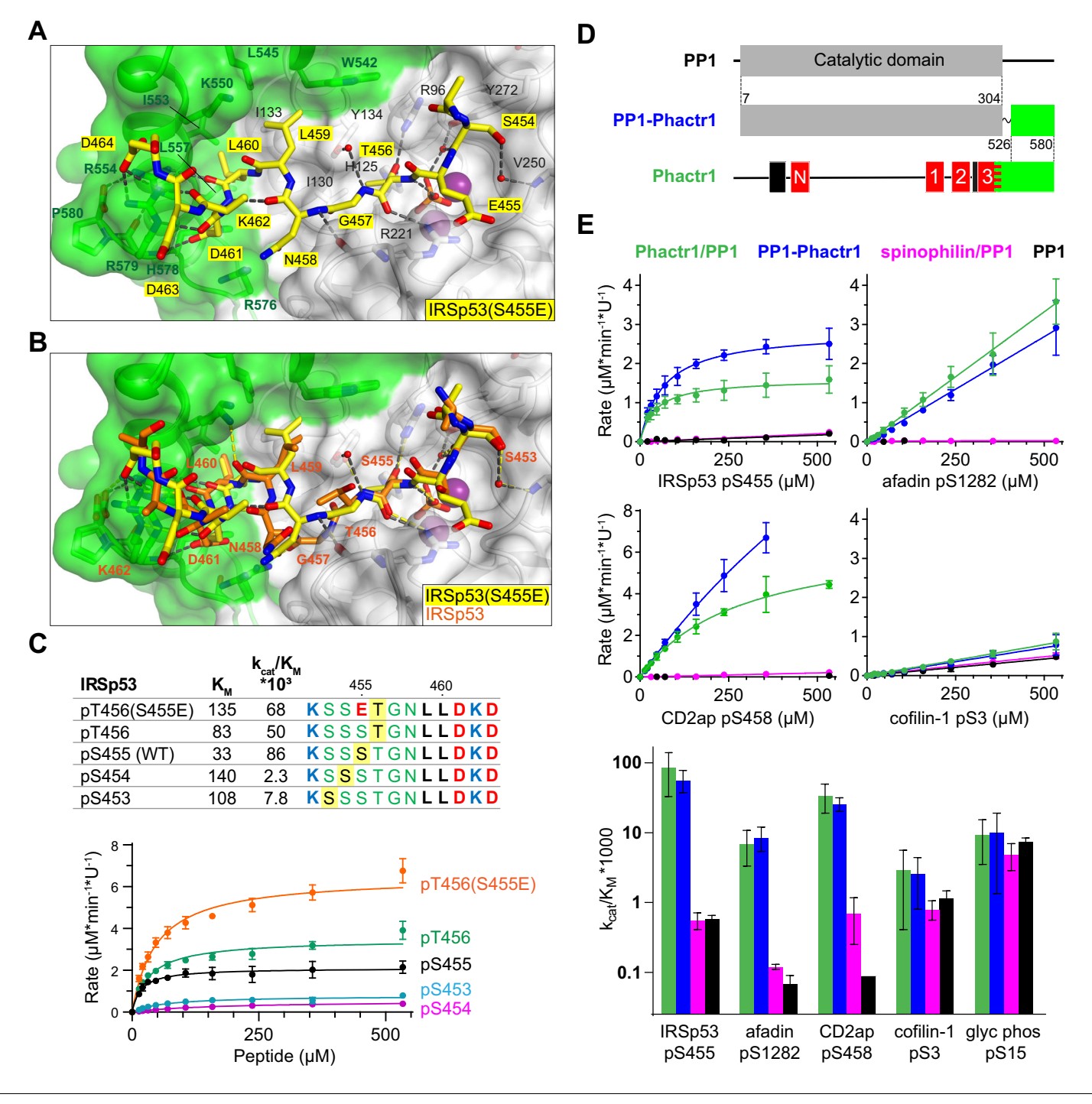

**Figure 6.** Flexible substrate interactions and substrate specificity of the Phactr1/PP1 complex. (**A, B**) Flexibility in target serine-hydrophobic pocket binding residue spacing, illustrated by (**A**) structure of the Phactr1/PP1-IRSp53(S455E) complex, compared with (**B**) the Phactr1/PP1-IRSp53 wildtype complex. (**C**) Phactr1/PP1 dephosphorylation of derivatives of IRSp53 peptides carrying phosphate at different locations, highlighted in yellow. Phosphatase activity data is shown below (data are mean ± SD, WT n = 15, others n = 2–6) (**D**) Schematic of the PP1-Phactr1 fusion protein PP1-Phactr1 (526-580). (**E**) Top, phosphatase activity data for the indicated substrates and enzymes. Bottom, relative catalytic efficiencies for the different substrates (data are mean ± SD, n = from 1 to 15).

The online version of this article includes the following figure supplement(s) for figure 6:

**Figure supplement 1.** Flexible substrate interactions and substrate specificity of the Phactr1/PP1 complex.

## Discussion

HeWe have elucidated the mechanism by which interaction of the RPEL protein Phactr1 with PP1 defines the substrate specificity of the Phactr1/PP1 holoenzyme. The Phactr1 C-terminal residues remodel the PP1 hydrophobic groove to form a new composite hydrophobic pocket and an adjacent amphipathic cavity, topped by a basic rim. We identified Phactr1/PP1 substrates, and showed that they make extensive contacts with the remodelled PP1 hydrophobic groove that are required for efficient catalysis. These contacts impose constraints on the substrate primary sequence, which allow the definition of a core Phactr1/PP1 dephosphorylation motif. The sequence conservation between Phactr family members suggests that the other Phactr-family/PP1 holoenzymes will have similar specificity. The substrates exhibit >100 fold enhanced reactivity with Phactr1/PP1, when compared with apo-PP1, or the spinophilin/PP1 complex, whose remodelled hydrophobic groove is different. Our results shed new light onto the role of PIPs in substrate recognition by PPP-family phosphatase complexes.

Phactr1 embraces PP1 using an extended RVxF-ϕϕ-R-W sequence string used by several other PIPs. We exploited the close approach of the Phactr1 sequences to the PP1 C-terminus to generate a PP1-Phactr1 fusion, which has a similar specificity to intact Phactr1/PP1 holoenzyme. This approach is likely to be applicable to PNUTS, PPP1R15A/B and spinophilin/neurabin, since these proteins all use an RVxF-ϕϕ-R-W string to bind PP1 in a strikingly similar way to Phactr1. The overlap of the RVxF motif with RPEL3 explains why G-actin competes with PP1 to bind Phactr1 and Phactr4, allowing the formation of the Phactr1/PP1 complex to be controlled by fluctuations in actin dynamics (*Huet et al., 2013*; *Wiezlak et al., 2012*). Although both G-actin and PP1 bind Phactr1 with nanomolar affinity, their binding sites are overlapping rather than superposable, which should facilitate their exchange.

## Phactr1/PP1 substrates include many cytoskeletal proteins and regulators

We used proteomic approaches coupled with Phactr1 overexpression and inactivation studies to show that Phactr1/PP1 dephosphorylates multiple target proteins involved in cytoskeletal structures or regulation. This supports previous studies showing that cytoskeletal phenotypes result from Phactr1 and Phactr4 mutations (*Hamada et al., 2018*; *Kim et al., 2007*; *Zhang et al., 2012*), and are induced by overexpression of Phactr1 and Phactr4 derivatives that constitutively associate with PP1 (*Huet et al., 2013*; *Wiezlak et al., 2012*). Inspection of the top-scoring Phactr1/PP1 substrates shows that acidic residues are over-represented C-terminal to the target phosphorylation site, and that hydrophobic residues are preferred at positions +4/+5.

Signal-regulated dephosphorylation of actin regulators by Phactr/PP1 complexes provides a new perspective on cytoskeletal regulation. For example IRSp53 was previously characterised as a Cdc42/Rac effector that controls F-actin assembly at membrane protrusions (*Scita et al., 2008*). The Phactr1/PP1 target residue pS455 characterised here is one of four negatively-acting putative AMPK sites that recruit 14-3-3 proteins (*Cohen et al., 2011*; *Kast and Dominguez, 2019*; *Robens et al., 2010*) and our data indicate another of these sites, S367, is also a Phactr1/PP1 target (*Supplementary file 1A*). Rho-actin signalling thus provides an additional positively acting signal input to IRSp53, probably controlling interaction of the neighbouring SH3 domain with its effectors. Interestingly, the Phactr1/PP1 sites in spectrin αII and girdin are also adjacent to SH3 and SH2 domains, respectively (*Lin et al., 2014*; *Figure 3C*; *Supplementary file 1*). Our data also confirm that the actin depolymerising factors cofilin and destrin are also dephosphorylated (and activated) by Phactr1, in agreement with previous studies of Phactr4 (*Huet et al., 2013*). Since Phactr proteins are inactivated by G-actin, their activation of cofilin potentially provides a feedback loop that couples F-actin severing to decreased G-actin level, but more work is required to establish this.

Phactr1 is enriched at the post-synaptic density (PSD) (*Allen et al., 2004*), and is required for neuronal migration and arborisation (*Hamada et al., 2018*). We identified multiple Phactr1-dependent substrates in hippocampal neurons. Several of these, including IRSp53 and spectrin αII, are dephosphorylated during long-term potentiation (LTP) (*Li et al., 2016*), and IRSp53-null mice exhibit deficits in hippocampal learning and memory (*Bobsin and Kreienkamp, 2016*; *Kang et al., 2016*). Moreover, dephosphorylation of cofilin is implicated in early-stage dendritic spine remodelling, along with G-actin itself (*Bosch et al., 2014*; *Lei et al., 2017*). These data suggest that rho-actin signalling

to Phactr1 and the resulting protein dephosphorylations may contribute to synaptic plasticity, and indeed in humans Phactr1 mutations cause the infantile seizure condition West syndrome (*Hamada et al., 2018*). There may be multiple targets for rho-actin signalling to RPEL proteins in this setting, as the RPEL protein ArhGAP12 (*Diring et al., 2019*) also influences dendritic spine morphology (*Ba et al., 2016*). Future work will focus on the functional significance of neuronal rho-actin signalling to Phactr/PP1 substrates.

## Determinants of Phactr1/PP1 substrate recognition

We used a PP1-substrate peptide fusion strategy to characterise Phactr1/PP1-substrate interactions. In our structures, the substrate sequences are not phosphorylated, but a phosphate is recruited to the catalytic cleft: they thus appear to represent enzyme/product complexes, presumably stabilised by virtue of the protein fusion. The inversion of the phosphate relative to its orientation in the Phactr1/PP1 holoenzyme provides persuasive support for the in-line nucleophilic attack model for PPP phosphatases proposed by others (*Egloff et al., 1995*; *Swingle et al., 2004*). Substrate interaction with the catalytic cleft, which does not change its conformation, is predominantly mediated by mainchain interactions with residues conserved among the PPP family. We were surprised to see that Phactr1/PP1 substrates dock with the catalytic cleft in the opposite orientation to that previously seen in a complex between PP5 and a phosphomimetic substrate derivative (*Oberoi et al., 2016*). However, the clear sequence bias observed in Phactr1/PP1 substrate sequences suggests that the orientation seen in our IRSp53 and spectrin αII complexes is strongly preferred.

The most striking aspect of substrate recognition is the role played by the composite surface generated by the close apposition of the Phactr1 extreme C-terminal sequences and the PP1 hydrophobic groove. This creates a new hydrophobic pocket, into which the substrate +4/+5 hydrophobic residues are inserted, and an adjacent amphipathic cavity, which appears less important, making no specific contacts with IRSp53, and being only partially occupied by spectrin αII. The preference for acidic residues C-terminal to the dephosphorylation site presumably reflects the basic electrostatics associated with the novel composite Phactr1/PP1 surface. Biochemical studies suggest that the additional binding energy provided by substrate interactions with the hydrophobic pocket is responsible for the enhanced catalytic rate of Phactr1/PP1 compared with apo-PP1 or other PP1 complexes. According to this view, it might be expected that substrates with higher $K_M$ might be more promiscuous in their interaction with PP1 complexes. Cofilin pS3 might be such a substrate: it exhibits similar catalytic efficiency to afadin pS1275, but has little preference for Phactr1/PP1 over apo-PP1 or spinophilin/PP1. However, high binding affinities might limit catalytic efficiency by slowing product dissociation, a situation perhaps exemplified by spectrin. PP1 inhibitors generally lack specificity because they target the catalytic site (*Zhang et al., 2013*), and our findings suggest that targeting the composite Phactr1/PP1 surface may be a good strategy to create specific Phactr/PP1 inhibitors.

An interesting consequence of the winding trajectory of the substrate following the catalytic site, and its strong interaction with the hydrophobic pocket, is that variation in the number of residues between the two can be tolerated. For example, IRSp53 pT456, with only four residues to the pocket-bound L460, has comparable kinetic properties to the bona fide IRSp53 pS455. We speculate that Phactr1/PP1 substrates such as afadin pS1275, where the sole hydrophobic residue is at +4 relative to the target residue, may well represent such 'stretched' substrates. This flexibility complicates the unambiguous definition of the dephosphorylation consensus site, and further studies will be required to produce a structure-based alignment of the substrates that we have identified.

We propose that Phactr/PP1 substrates can be defined by a hydrophobic doublet at position +4/+5 or +3/+4 relative to the dephosphorylation site, with leucine preferred at the distal position, embedded within acidic sequences. The simple Phactr1/PP1 core recognition sequence, S/T-x$_{(2-3)}$-φ-L, is reminiscent of those seen for protein kinases (*Miller and Turk, 2018*) and is perhaps the first identified for PP1. Mutation of positions +4 and/or +5 in candidate Phactr/PP1 substrates will be a useful way to generate authentic constitutively phosphorylated mutants for functional studies, as an alternative to 'phosphomimetic' glutamate substitution of the phosphorylated residue.

## Comparison with other substrate-specificity PIPs

The lack of structural information for PP1-substrate complexes has hitherto precluded demonstration of how PIP-dependent remodelling of the substrate-binding grooves of PP1 can influence its ability

to recognise substrate. In spinophilin, which also binds PP1 via a RVxF-ϕϕ-R-W string, sequences C-terminal to the W-motif form a helix that remodels the PP1 hydrophobic groove differently (*Ragusa et al., 2010*). Spinophilin therefore does not detectably enhance PP1 activity towards Phactr/PP1 substrates, although the relative contributions of the spinophilin-remodelled PP1 surface, and the adjacent PDZ domain, to substrate specificity remain to be determined (*Ragusa et al., 2010*). In contrast, MYPT interacts with PP1 in a different way to Phactr1 and spinophilin, extending both the C-terminal and acidic grooves (*Terrak et al., 2004*), and it will be interesting to see how this facilitates substrate recognition. In all these complexes, PIP-PP1 interaction also acts indirectly to constrain substrate binding specificity, by occluding potential substrate-binding sites on the PP1 surface such as the RVxF binding pocket (*Ragusa et al., 2010*).

Our results underscore the importance of the composite Phactr1/PP1 surface in substrate recognition and specificity. Nevertheless, as with protein kinases, these interactions alone are unlikely to define substrate selection completely (*Miller and Turk, 2018*). Phactr-family members do not appear to contain any conserved domains that might represent autonomous substrate-binding domains, as seen in NIPP1 (*Boudrez et al., 2000*) and PPP1R15A (*Chen et al., 2015*; *Choy et al., 2015*; *Crespillo-Casado et al., 2018*). On the other hand, many PIPs act to target PP1 complexes to specific subcellular compartments, proteins or macromolecules (*Cohen, 2002*). Phactr-family proteins exhibit differential subcellular localisations, including the nucleus and cell membrane (*Huet et al., 2013*; *Wiezlak et al., 2012*), and Phactr1 has been shown to interact directly with the KCNT1 potassium channel (*Ali et al., 2020*; *Fleming et al., 2016*). At least for Phactr3 and Phactr4, membrane targeting involves conserved N-terminal sequences (*Huet et al., 2013*; *Itoh et al., 2014*) which overlap a G-actin controlled nuclear import signal in Phactr1. The genetic dominance of West syndrome Phactr1 mutations and Phactr4 R536P/humdy mutation also is consistent with Phactr-family proteins interacting with other cellular components in addition to PP1 (*Hamada et al., 2018*; *Kim et al., 2007*; *Zhang et al., 2012*). These considerations, and the coupling of Phactr1/PP1 holoenzyme formation to rho-actin signalling, make it likely that the range of substrates controlled by Phactr1/PP1 in a particular setting will reflect the state of cellular actin dynamics and subcellular localisation of the complexes as well as the direct recognition of substrate primary sequence by the holoenzyme.

## Materials and methods

### Plasmids

pET28 PP1(7–330) and pcDNA3.1 IRSp53 were gifts from Wolfgang Peti and Eunjoon Kim (KAIST, S. Korea) respectively. Other plasmids were: modified pTRIPZ (*Diring et al., 2019*); pEF Phactr1 and derivatives (*Wiezlak et al., 2012*); pGEX 6p2 (GE Healthcare); and pGRO7 (Takara). For protein expression, Phactr1(507-580) and Phactr1(516-580) sequences were expressed using pGEX-6P2. PP1-substrate chimeras were derivatives of pET28 PP1(7–330) in which PP1 sequences 7–304 were joined by a (Ser-Gly)$_9$ linker to either IRSp53(449-465) (QQGKSSSTGNLLDKDDL) IRSp53(449-465)-S455E (QQGKSS**E**TGNLLDKDDL), or spectrin αII (1025–1039) (DPAQSASRENLLEEQ). pET28 PP1-Phactr1(526-580), derived from pET28 PP1(7–330), encodes PP1(7–304)-SGSGS-Phactr1(526-580).

Plasmid construction and mutagenesis used standard methods, the NEB NEBuilder HiFi DNA Assembly Cloning Kit, or the NEB Q5 Site-Directed Mutagenesis Kit. Primers are listed in *Supplementary file 4*.

### Protein expression and purification

Protein expression in BL21 (DE3) *E. coli* cells (Invitrogen) was with pGRO7 coexpression as described (*Choy et al., 2014*). Overnight pre-cultures (400 ml) were grown in LB medium supplemented 1 mM MnCl$_2$ and used to inoculate a 100L fermenter. After growth to OD$_{600}$ of ~0.5, 2 g/L of arabinose was added to induce GroEL/GroES expression. At OD$_{600}$ ~1, the temperature was lowered to 10℃ and protein expression induced with 0.1 mM IPTG for ~18 hr. Cells were harvested, re-suspended in fresh LB medium/1 mM MnCl$_2$/200 µg/ml chloramphenicol and agitated for 2 hr at 10℃. Harvested cells were resuspended in lysis buffer (50 mM Tris-HCl, pH 8.5, 5 mM imidazole, 700 mM NaCl, 1 mM MnCl$_2$, 0.1% v/v TX-100, 0.5 mM TCEP, 0.5 mM AEBSF, 15 µg/ml benzamidine and complete EDTA-free protease inhibitor tablets), lysed by French press, clarified, and stored at −80℃. Phactr1/PP1 complexes were batch-adsorbed onto glutathione-sepharose affinity matrix, and recovered by

cleavage with 3C protease at 4°C overnight in 50 mM Tris-HCl, pH 8.5, 500 mM NaCl, 0.5 mM TCEP. Eluted complex was further purified via adsorption on Ni-NTA IMAC, and elution with 50 mM Tris pH 8.0, 200 mM Imidazole, 700 mM NaCl and 1 mM MnCl$_2$ at 4°C. Finally, proteins were purified using size exclusion chromatography on a Superdex 200 26/60 column equilibrated in complex buffer (20 mM Tris-HCl pH 8.5, 0.25 M NaCl and 0.4 mM TCEP).

## Bio-layer interferometry

Bio-layer interferometry (BLI) was as described (*Bertran et al., 2019*), using the Octet Red 96 (Forte-Bio). 50 µg/ml His-tagged PP1α was immobilised on Nickel-coated biosensor (Ni-NTA, ForteBio), and the loaded biosensors then incubated with 0.1–10 µM Phactr1 peptides in Octet buffer (50 mM Tris pH 7.5, 500 mM NaCl, 0.5 mM TCEP, 0.1% Tween 20, 500 mg BSA/100 ml). Curve fitting, steady state analysis, and calculation of kinetic parameters were done using Octet software version 7.1 (ForteBio). For peptides used, see *Supplementary file 4*.

## Crystallisation and structure determination

Phactr1/PP1 in complex buffer was concentrated to 10 mg/ml and crystallised at 20°C using sitting-drop vapour diffusion. Sitting drops of 1 µl consisted of a 1:1 (vol:vol) mixture of protein and well solution. Well solutions were as follows. Phactr(507-580)/PP1α(7-300): 7.5% PEG 3350, 0.2 M MgSO$_4$; Phactr1(516-580)/PP1α(7-300): 1M LiCl, 0.1 M tri-sodium citrate pH 5.25, 10% PEG 6000; Phactr1/PP1-IRSp53(S455E): 20% PEG 3350, 0.2 M NaBr; PP1-Phactr1(526-580), 20% PEG 3350, 0.2M Potassium Citrate. Crystals appeared within 24–48 hr and reached their maximum size after 4 to 7 days, apart from Phactr(507-580)/PP1α(7-300), for which the best crystals appeared after 3–7 weeks, reaching their maximum size after 8 weeks. For Phactr1/PP1-IRSp53 and Phactr1/PP1-spectrin α II complexes, crystallisation was achieved by microseed matrix screening (*D'Arcy et al., 2014*) using Phactr1/PP1-IRSp53(S455E) crystals. Phactr1/PP1-IRSp53 crystallised in 20% PEG 3350, 0.2 M KSCN,0.1 M BIS-Tris propane pH 8.5; Phactr1/PP1-spectrin αII crystallised in 20% PEG 3350, 0.2 M NaI, 0.1 M BIS-Tris propane pH 8.5. Crystals appeared within a day and reached maximum size within 4 to 5 days.

All crystals were cryoprotected in well solution supplemented with 15% glycerol + 15% ethylene glycol and flash-frozen in liquid nitrogen. X-ray data were collected at 100 K at beamlines I04-1 (mx9826-17), I02 (mx9826-26), I03 (mx18566-37), I24 (mx18566-38), I04 (mx18566-29), and I04-1 (mx18566-55) of the Diamond Light Source Synchrotron (Oxford, UK). Data collection and refinement statistics are summarised in *Table 1*. Data sets were indexed, scaled, and merged with xia2 (*Winter et al., 2013*). Molecular replacement used the atomic coordinates of human PP1 from PDB 4M0V (*Choy et al., 2014*) in PHASER (*McCoy et al., 2007*). Refinement used Phenix (*Adams et al., 2010*). Model building used COOT (*Emsley et al., 2010*) with validation by PROCHECK (*Vaguine et al., 1999*). Residues modelled in the different structures are summarised in *Supplementary file 5*. Figures were prepared using PYMOL (*Schrodinger LLC, 2020*).

Atomic coordinates and crystallographic structure factors have been deposited in the Protein Data Bank under accession codes PDB 6ZEE, Phactr(507-580)/PP1α(7-300); PDB 6ZEF, Phactr1(516-580)/PP1α(7-300); PDB 6ZEG, Phactr1/PP1-IRSp53; PDB 6ZEH, Phactr1/PP1-spectrin; PDB 6ZEI, Phactr1/PP1-IRSp53(S455E); PDB 6ZEJ, PP1(7–304)-SGSGS-Phactr1(526-580).

## Phosphatase assays

Phosphatase assays (50 µl) were performed in 96-well plates. Peptides (40 µl) were serially diluted 1.5-fold from 1 mM in complex buffer (20 mM Tris-HCl pH 8.5, 250 mM NaCl, 0.4 mM TCEP). Phactr1/PP1 (0.2–3U, 10 µl) was added and after 15 min at room temperature, reactions were quenched by addition of 100 µl of Biomol Green reagent (Enzo Life Sciences) for 30 min, Absorbances at 620 nm were measured using the SpectraMax Plus 384 microplate reader and converted into phosphate concentrations using standard curves. Rate constants were estimated in GraphPad Prizm 8 by fitting product concentration readouts to modified Michaelis-Menten equation:

$$\frac{P}{t*E} = \frac{k_{cat}}{K_M} * \frac{C}{\left(\frac{C}{K_M} + 1\right)}$$

(P, product released at time t; E, Phactr1/PP1 concentration; $k_{cat}/K_M$, catalytic efficiency; C, initial substrate concentration; $K_M$, Michaelis constant).

The activity of the Phactr1/PP1 preparation was established on the day of each experiment, using 125 µM IRSp53(447-465) pS455 peptide as standard, with Phactr1/PP1 at various concentrations. One unit of phosphatase activity was defined as the concentration of Phactr1/PP1 complex that generates 15 µM of phosphate in 15 min. To normalise the activity of different phosphatases, a para-nitrophenylphosphate (pNPP)-based assay was used. 10 µl phosphatase (200 nM) was added to 40 µl pNPP (2-fold serial dilution from 100 mM), incubated at room temperature for 15 min, then quenched with 25 µl 3 M NaOH. Para-nitrophenol product was measured by absorbance at 405 nm. Full assay data can be found in *Supplementary file 2*.

## Peptides and peptide array binding assays

Peptides were synthesised by the Francis Crick Institute Peptide Chemistry Science Technology Platform using standard techniques. Peptide arrays (5–10 nmol/spot) were synthesised on a derivatised cellulose membrane Amino-PEG$_{500}$-UC540 using Intavis ResPep SLi automated synthesiser (Intavis Bioanalytical Instruments).

Dry membranes were blocked for 1 hr in 5% milk in Tris-buffered saline supplemented with 0.1% Tween-20 (TBST) with agitation, rinsed with TBST, and incubated overnight at 4°C with 10 µg/ml GST-Phactr1(516-580)/PP1(7–300) complex in TBST. After three 10-min washes with TBST, membranes were incubated with 1:5000 HRP-conjugated anti-GST in 5% milk/TBST at room temperature for 1 hr, washed three times with TBST, and binding revealed using SuperSignal West Pico Plus reagent (ThermoFisher) as chemiluminescent substrate. Images were taken using Amersham Imager 600 (GE).

## Cells, transfection and RNA analysis

Mouse NIH3T3 fibroblasts (mycoplasma-free) were maintained in DMEM (Gibco) with 10% fetal calf serum (FCS) and penicillin-streptomycin at 37°C and 10% $CO_2$. Cells were transfected using Lipofectamine 2000 (Invitrogen) (150,000 cells per well, six-well dish). For SILAC proteomics, NIH3T3 cell line pools were generated stably carrying doxycyline-inducible pTRIPZ-Phactr1 derivatives, using puromycin selection. Phactr1 expression was induced by doxycycline addition as indicated in the figure legends. Cells were maintained overnight in DMEM/0.3% FCS, and then stimulated with 15% FCS for 1 hr or cytochalasin D (CD) or latrunculin B (LatB) for 30 min. Primary rat hippocampal neurons (DIV 14) were cultured as described (*Baltussen et al., 2018*) and transfected using Lipofectamine 2000.

Total RNA was purified using the GenElute mammalian total RNA kit (Sigma) and cDNA synthesised using the Transcriptor First Strand cDNA Synthesis kit (Roche) with random hexamer primers. Real-time-qPCR was performed using the 7500 Fast Real-Time PCR System (Thermo Fisher Scientific) with the SYBR green reaction mix (Life Technologies). Primers are listed in *Supplementary file 4*. The relative abundance of target cDNA was normalised against rps16 cDNA abundance in each sample.

## Immunofluorescence and immunoblotting

Immunofluorescence microscopy in fibroblasts was performed as described (*Vartiainen et al., 2007*; *Wiezlak et al., 2012*). F-actin was detected with FITC-phalloidin (Invitrogen) and nuclei were visualised using DAPI. For immunofluorescence microscopy primary rat neurons were transfected with expression plasmids and fixed 24 hr later in 4% paraformaldehyde/4% sucrose in PBS before staining with Flag and GFP antibodies. Images were taken with Leica SP5 confocal microscope with 63x (NA 1.4) oil objective. GFP was used to monitor the shape of dendritic spines, and Flag for transfected cells. An image stack with 0.5 µm z-intervals was obtained to capture a part dendritic arbour at high magnification. In each experiment, at least 10 cells per mutant were imaged and the morphology of dendritic spines was quantified blindly. Statistical analysis was performed in GraphPad Prizm eight using Welch's t-test function.

SDS-PAGE analysis of cell lysates and immunoblotting was performed using standard techniques; the signal was visualised and quantified using Odyssey CLx instrument (LI-COR) and the Image Studio (LI-COR) Odyssey Analysis Software.

Primary antibodies used were anti-Flag (Sigma, F7425), anti-IRSp53 (Santa Cruz, sc-50011 and Abcam, ab15697), anti-afadin (Santa Cruz, sc-74433), anti-GST (VWR, RPN1236). Secondary antibodies labelled with IRDye 800CW and IRDye 680LT were from LICOR. Rabbit anti-IRSp53 pS455, anti-afadin pS1275 and anti-spectrin pS1031 antibodies were custom-made (Covalab); for antigen sequences used see *Supplementary file 4*.

## SILAC phosphoproteomics in NIH3T3 cells

Cells were maintained for at least six passages in 'heavy' (R10K8) or 'light' (R0K0) DMEM medium supplemented with 10% SILAC dialysed fetal calf serum (3 kDa cutoff). Phactr1 expression was induced by doxycycline addition to 1 µg/ml for 5 hr. Cells were harvested processed for mass spectrometry essentially as described (*Pattison et al., 2016*; *Touati et al., 2018*) with minor modifications. 1 mg each of 'light' and 'heavy' labelled lysates were mixed and dried. Protease digestion and phosphopeptide enrichment was done as described with minor modifications (*Pattison et al., 2016*; *Touati et al., 2018*). Peptides were dissolved in 35 µl of 1% TFA, with sonication, and fractionated on a 50 cm, 75 µm I.D. Pepmap column with elution directly into the LTQ-Orbitrap Velos. Xcalibur software was used to setup data-dependent acquisition in top10 mode. Raw mass spectrometric data were processed in MaxQuant (version 1.3.0.5) for peptide and protein identification; database search was with the Andromeda search engine against the *Mus musculus* canonical sequences from UniProtKB. Fixed modifications were set as Carbamidomethyl (C) and variable modifications set as Oxidation (M), Acetyl (Protein N-term) and Phospho (STY). The estimated false discovery rate was set to 1% at the peptide, protein, and site levels. A maximum of two missed cleavages were allowed. Phosphorylation site tables were imported into Perseus (v1.4.0.2) for analysis. Contaminants and reverse peptides were cleaned up from the Phosphosites (STY).

Dephosphorylation score was defined as $\log_2(\text{dephosphorylation score}) = 0.25*[\log_2(2^H/1^L) + \log_2(3^H/1^L) + \log_2(2^L/1^H) + \log_2(3^L/1^H)]$ where 1,2 and 3 denote Phactr1$^{XXX}$, Phactr1$^{XXX}\Delta$C, and empty vector cells, and H and L denote samples from cells grown in R10K8 and R0K0 media, respectively. Sequence logos were generated using WebLogo (https://weblogo.berkeley.edu/logo.cgi/).

The mass spectrometry proteomics data have been deposited to the ProteomeXchange Consortium (http://proteomecentral.proteomexchange.org) via the PRIDE partner repository with the dataset identifier PXD019977.

## TMT phosphoproteomics in neurons

The Phactr1-null (tm1d) allele was derived from the CSD79794 Phactr1 Tm1a allele (KOMP; https://www.komp.org/geneinfo.php?geneid=74176) by sequential action of Flp and Cre. Heterozygous animals were crossed. Hippocampal and cortical tissue was extracted from E16.5 embryos, and cultured in 12-well plate dishes (500,000 cells per well) as described (*Baltussen et al., 2018*) before genotyping. Two biological replicates were processed for wildtype or Phactr1-null neurons. On DIV10, neurons were treated for 30 min with CD (10 µM), LB (1 µM) or vehicle (DMSO). Preparation of lysates, protease digestion and phosphopeptide enrichment was done as described with minor modifications (*Eder et al., 2020*). Phosphopeptides were eluted directly into the Orbitrap Fusion Lumos, operated with Xcalibur software, with measurement in MS2 and MS3 modes. The instrument was set up in data-dependent acquisition mode, with top 10 most abundant peptides selected for MS/MS by HCD fragmentation.

Raw mass spectrometric data were processed in MaxQuant (version 1.6.2.10); database search against the *Mus musculus* canonical sequences from UniProtKB was performed using the Andromeda search engine. Fixed modifications were set as Carbamidomethyl (C) and variable modifications set as Oxidation (M), Acetyl (Protein N-term) and Phospho (STY). The estimated false discovery rate was set to 1% at the peptide, protein, and site levels, with a maximum of two missed cleavages allowed. Reporter ion MS2 or Reporter ion MS3 was appropriately selected for each raw file.

Phosphorylation site tables were imported into Perseus (v1.6.1.2) for analysis. Contaminants and reverse peptides were cleaned up from the Phosphosites (STY) and the values normalised using Z-score function across columns.

Cortical and hippocampal as well as MS2/MS3 data across two biological replicates were pooled. (DMSO-CD) differences were calculated and compared between Phactr1 WT and KO neurons using the two-sample t-test. Phosphorylation sites exhibiting significantly different dephosphorylation in

WT neurons compared with KO neurons were considered to be Phactr1-dependent. Sequence logos were generated using WebLogo (https://weblogo.berkeley.edu/logo.cgi/).

The mass spectrometry proteomics data have been deposited to the ProteomeXchange Consortium (http://proteomecentral.proteomexchange.org) via the PRIDE partner repository with the dataset identifier PXD019882.

## Acknowledgements

We dedicate this paper to the memory of Tricia Cohen, who founded the field of PPP-family molecular biology. We thank the Crick Science Technology platforms for support and advice during this work, Clare Watkins and Julie Bee (Biological Resources), Namita Patel, Damini Patel and Alireza Alidoust (Fermentation Facility), Graham Clark (Genomics Equipment Park), Kurt Anderson (Light Microscopy) and Phil Walker and Andrew Purkiss (Structural biology). X-ray data were collected at the Diamond Light Source on beamlines I04-1 (mx9826-17), I02 beamline (mx9826-26), I03 (mx18566-37), I24 (mx18566-38), I04 (mx18566-29) and I04-1 (mx18566-55). We thank Wolfgang Peti (Arizona University) and Eunjoon Kim (KAIST) for the spinophilin and IRSp53 expression plasmids respectively, and Michael Way, Neil McDonald, Peter Parker, Nic Tapon and members of the RT and SU groups for helpful discussions throughout the course of this project and/or comments on the manuscript. This work was supported by ERC Advanced Grant 268690 to RT; by Cancer Research UK core funding until March 31 st 2015; and since 2015 by the Francis Crick Institute, which receives its core funding from Cancer Research UK (FC001-190 to RT, FC001-201 to SU), the UK Medical Research Council (FC001-190 to RT, FC001-201 to SU) and the Wellcome Trust (FC001-190 to RT, FC001-201 to SU).

## Additional information

### Funding

| Funder | Grant reference number | Author |
| --- | --- | --- |
| H2020 European Research Council | 268690 | Richard Treisman |
| Cancer Research UK | FC001-190 | Richard Treisman |
| Medical Research Council | FC001-190 | Richard Treisman |
| Wellcome Trust | FC001-190 | Richard Treisman |
| Cancer Research UK | FC001-201 | Sila Ultanir |
| Medical Research Council | FC001-201 | Sila Ultanir |
| Wellcome Trust | FC001-201 | Sila Ultanir |

The funders had no role in study design, data collection and interpretation, or the decision to submit the work for publication.

### Author contributions

Roman O Fedoryshchak, Data curation, Formal analysis, Investigation, Visualization, Methodology, Writing - original draft, Writing - review and editing, identified neuronal Phactr1 substrates, conducted neuron transfections, analysed proteomic data, conducted the enzymology and substrate binding studies, generated phospho-specific antisera, and devised the PP1-Phactr1 fusion protein and dephosphorylation resistant IRSp53 mutants; Magdalena Přechová, Formal analysis, Investigation, Visualization, Methodology, Writing - original draft, Writing - review and editing, characterised the Phactr1/PP1 complex, identified and characterised fibroblast Phactr1/PP1 substrates, generated phospho-specific antisera, generated the Phactr1-null mouse, and conducted neuron transfections; Abbey M Butler, Rebecca Lee, Formal analysis, Investigation, Writing - review and editing, expressed and purified proteins and conducted crystallisations; Nicola O'Reilly, Resources, Methodology, Writing - review and editing, synthesised peptides and arrays; Helen R Flynn, Formal analysis, Methodology, Writing - review and editing, conducted proteomic analysis; Ambrosius P Snijders, Supervision,

Methodology, Writing - review and editing, conducted proteomic analysis; Noreen Eder, Investigation, Methodology, Writing - review and editing, cultured primary neurons and conducted proteomic analysis; Sila Ultanir, Resources, Methodology, Writing - review and editing, cultured primary neurons and conducted proteomic analysis; Stephane Mouilleron, Conceptualization, Data curation, Formal analysis, Supervision, Investigation, Visualization, Methodology, Writing - original draft, Writing - review and editing, conducted biophysical studies, devised the substrate fusion strategy, and determined the structures; Richard Treisman, Conceptualization, Resources, Formal analysis, Supervision, Funding acquisition, Methodology, Writing - original draft, Writing - review and editing, conceived the project, designed and interpreted experiments, and wrote the paper with input from other authors

### Author ORCIDs
Roman O Fedoryshchak ⓘ https://orcid.org/0000-0003-1865-8372
Magdalena Přechová ⓘ https://orcid.org/0000-0002-4591-854X
Abbey M Butler ⓘ https://orcid.org/0000-0003-0650-2100
Nicola O'Reilly ⓘ https://orcid.org/0000-0002-4204-6491
Helen R Flynn ⓘ https://orcid.org/0000-0001-7002-9130
Ambrosius P Snijders ⓘ https://orcid.org/0000-0002-5416-8592
Sila Ultanir ⓘ https://orcid.org/0000-0001-5745-3957
Stephane Mouilleron ⓘ https://orcid.org/0000-0001-7977-6298
Richard Treisman ⓘ https://orcid.org/0000-0002-9658-0067

### Ethics
Animal experimentation: Animal experimentation complied with all ethical regulations and was carried out under the UK Home Office Project licence P7C307997 in the Crick Biological Research Facilities.

### Decision letter and Author response
Decision letter https://doi.org/10.7554/eLife.61509.sa1
Author response https://doi.org/10.7554/eLife.61509.sa2

## Additional files
### Supplementary files
• Supplementary file 1. Phosphoproteomics data. (A, B) SILAC phosphoproteomics in NIH3T3 cells. (A) NIH3T3 cells expressing doxycycline-inducible Phactr1$^{XXX}$, Phactr1$^{XXX}\Delta$C or vector alone were cultured in R0K0 or R10K8 SILAC media, protein expression induced with doxycycline, and cell lysates analysed by MS proteomics. Phosphorylation sites are annotated with dephosphorylation score, raw H/L values listed. (B) 1D enrichment analysis of Gene Ontology terms and kinase motifs based on the dataset from A, table A. Terms with Benjamini-Hochberg FDR < 0.02 are shown. Gene Ontology Biological Process terms are in bold and those with positive mean value are reported in *Figure 3B*. (C-E) Wildtype (WT) or Phactr1-null (KO) cortical and hippocampal neurons DIV10 were treated with DMSO vehicle, CD or LB for 30' and then analysed by TMT phosphoproteomics in MS2 and MS3 modes. Reporter ion intensities were normalised using Z-score function. (C) Raw and Z-score values for MS2 mode. (D) Raw and Z-score values for MS3 mode. (E) *t*-test was applied to (DMSO - CD) differences in WT neurons vs KO neurons. Phosphorylation sites are described and annotated with *t*-test difference and significance.

• Supplementary file 2. Phosphatase activity source data.

• Supplementary file 3. Immunoblot source data.

• Supplementary file 4. Peptides and oligonucleotides.

• Supplementary file 5. Summary of modelled residues for the different structures.

• Transparent reporting form

## Data availability

Diffraction data have been deposited in PDB under the accession code 6ZEE, 6ZEF, 6ZEG, 6ZEH, 6ZEI, 6ZEJ. The mass spectrometry proteomics data have been deposited to the ProteomeXchange Consortium (http://proteomecentral.proteomexchange.org) via the PRIDE partner repository with the dataset identifiers PXD019977 and PXD019882. All data generated or analysed during this study are included in the manuscript and supporting files.

The following datasets were generated:

| Author(s) | Year | Dataset title | Dataset URL | Database and Identifier |
|---|---|---|---|---|
| Přechová M, Mouilleron S, Treisman R, Fedoryshchak RO, Lee R, Butler A | 2020 | Structure of PP1-Phactr1 chimera [PP1(7-304) + linker (SGSGS) + Phactr1(526-580)] | https://www.rcsb.org/structure/6ZEJ | RCSB Protein Data Bank, 6ZEJ |
| Mouilleron S, Treisman R, Fedoryshchak RO, Lee R, Butler A, Přechová M | 2020 | Structure of PP1-IRSp53 S455E chimera [PP1(7-304) + linker (G/S)x9 + IRSp53(449-465)] bound to Phactr1 (516-580) | https://www.rcsb.org/structure/6ZEI | RCSB Protein Data Bank, 6ZEI |
| Přechová M, Mouilleron S, Fedoryshchak RO, Butler A, Lee R, Treisman R | 2020 | Structure of PP1(7-300) bound to Phactr1 (507-580) at pH8.4 | https://www.rcsb.org/structure/6ZEE | RCSB Protein Data Bank, 6ZEE |
| Přechová M, Mouilleron S, Fedoryshchak RO, Butler A, Lee R, Treisman R | 2020 | Structure of PP1(7-300) bound to Phactr1 (516-580) at pH 5.25 | https://www.rcsb.org/structure/6ZEF | RCSB Protein Data Bank, 6ZEF |
| Přechová M, Mouilleron S, Treisman R, Fedoryshchak RO, Butler A, Lee R | 2020 | Structure of PP1-IRSp53 chimera [PP1(7-304) + linker (G/S)x9 + IRSp53(449-465)] bound to Phactr1 (516-580) | https://www.rcsb.org/structure/6ZEG | RCSB Protein Data Bank, 6ZEG |
| Mouilleron S, Přechová M, Treisman R, Fedoryshchak RO, Lee R, Butler A | 2020 | Structure of PP1-spectrin alpha II chimera [PP1(7-304) + linker (G/S)x9 + spectrin alpha II (1025-1039)] bound to Phactr1 (516-580) | https://www.rcsb.org/structure/6ZEH | RCSB Protein Data Bank, 6ZEH |
| Přechová M, Fedoryshchak RO, Flynn HR, Snijders AP, Treisman R | 2020 | Phactr1/PP1 phosphatase substrates in mouse NIH3T3 fibroblasts | http://proteomecentral.proteomexchange.org/cgi/GetDataset?ID=PXD019977 | ProteomeXchange, PXD019977 |
| Fedoryshchak RO, Eder N, Flynn HR, Snijders AP, Ultanir S, Treisman R | 2020 | Phactr1/PP1 phosphatase substrates in mouse primary neurons | http://proteomecentral.proteomexchange.org/cgi/GetDataset?ID=PXD019882 | ProteomeXchange, PXD019882 |

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

# Appendix 1

## Key Resources Table

Appendix 1—key resources table

| Reagent type (species) or resource | Designation | Source or reference | Identifiers | Additional information |
|---|---|---|---|---|
| Gene *Mus musculus* | Afdn | UniProt | Q9QZQ1 | Encodes Afadin, also known as AF6, MLLT4 |
| Gene *Mus musculus* | Baiap2 | UniProt | Q8BKX1 | Encodes IRSp53, also known as BAIAP2 |
| Gene *Mus musculus* | Sptan1 | UniProt | P16546 | Encodes Spectrin, also known as SPTA2, Fodrin, SPTAN1 |
| Gene *Mus musculus* | Phactr1 | UniProt | Q2M3X8 | |
| Strain *E. coli* | 5-alpha competent *E. coli* | NEB | C2992I | |
| Strain *E. coli* | Protein expression BL21 (DE3) | NEB | C2527H | |
| Cell line *Mus musculus* | NIH3T3 | ATCC | CRL-1658 | |
| Strain, strain background *Mus musculus* | Phactr1^tm1d/tm1d | M. Prechova | | Constructed from Phactr1^tm1a(KOMP)Wtsi https://www.komp.org/ M. Přechová, PhD thesis (University College London) |
| Antibody | Rabbit polyclonal anti-Flag | Sigma | F7425, RRID:AB_439687 | WB 1:500 |
| Antibody | Goat polyclonal anti-IRSp53 | Abcam | ab15697 RRID:AB_302045 | WB 1:500 |
| Antibody | Mouse monoclonal anti-Afadin (B-5) | Santa Cruz | sc-74433 RRID:AB_1118734 | WB 1:200 |
| Antibody | anti-GST HRP Conjugate | VWR | RPN1236 RRID:AB_771429 | WB 1:1000 |
| Antibody | Mouse monoclonal anti-spectrin (D8B7) | Abcam | ab11755 RRID:AB_298540 | WB 1:500 |
| Antibody | Mouse monoclonal anti-Gapdh (G-9) | Santa-Cruz | sc-365062 RRID:AB_10847862 | WB 1:1000 |
| Antibody | Rabbit polyclonal anti-IRSp53 pS455 | This paper | | WB 1:500 See Materials and methods; *Figure 3F* |
| Antibody | Rabbit polyclonal anti-afadin pS1282 | This paper | | WB 1:500 See Materials and methods; *Figure 3F* |
| Antibody | Rabbit polyclonal anti-spectrin pS1031 | This paper | | WB 1:500 See Materials and methods; *Figure 3F* |
| Antibody | IRDye 680RD Secondary Antibodies | Licor | 925-68073 RRID:AB_2716687 | WB 1:10000 |

*Continued on next page*

*Appendix 1—key resources table continued*

| Reagent type (species) or resource | Designation | Source or reference | Identifiers | Additional information |
|---|---|---|---|---|
| Antibody | IRDye 800CW Secondary Antibodies | Licor | 925-32214 RRID:AB_2687553 | WB 1:10000 |
| Recombinant DNA reagent (plasmid) | pcDNA3.1 IRSp53 | Dr. Eunjoon Kim | PMID:15673667 | |
| Recombinant DNA reagent (plasmid) | pcDNA3.1 IRSp53 L460A | This paper | | See Materials and methods; *Figure 5B* |
| Recombinant DNA reagent (plasmid) | pEF Phactr1 | *Wiezlak et al., 2012* | | |
| Recombinant DNA reagent (plasmid) | pEF Phactr1(464-580) | This paper | | See Materials and methods; *Figure 3—figure supplement 1A* |
| Recombinant DNA reagent (plasmid) | pEF Phactr1$^{XXX}$(464-580) | This paper | | See Materials and methods; *Figure 3—figure supplement 1A* |
| Recombinant DNA reagent (plasmid) | pTRIPZ | *Diring et al., 2019* | | |
| Recombinant DNA reagent (plasmid) | pTRIPZ Phactr1$^{XXX}$ | This paper | | See *Wiezlak et al., 2012*; Materials and methods; *Figure 3—figure supplement 1B* |
| Recombinant DNA reagent (plasmid) | pTRIPZ Phactr1$^{XXX\Delta C}$ | This paper | | See *Wiezlak et al., 2012*; Materials and methods; *Figure 3—figure supplement 1B* |
| Recombinant DNA reagent (plasmid) | pGEX 6P2 | GE Healthcare | 27-4598-01 | |
| Recombinant DNA reagent (plasmid) | pGEX Phactr1(517-580) | This paper | | See Materials and methods; *Figure 1* |
| Recombinant DNA reagent (plasmid) | pGEX Phactr1(523-580) | This paper | | See Materials and methods *Figure 1* |
| Recombinant DNA reagent (plasmid) | pET28 PP1(7-300) | Dr. Wolfgang Peti | RRID:Addgene_26566 | |
| Recombinant DNA reagent (plasmid) | pET28 PP1-(SG)9-IRSp53 S455E | This paper | | See Materials and methods; *Figure 6* |

*Continued on next page*

*Appendix 1—key resources table continued*

| Reagent type (species) or resource | Designation | Source or reference | Identifiers | Additional information |
|---|---|---|---|---|
| Recombinant DNA reagent (plasmid) | pET28 PP1-(SG)9-IRSp53 | This paper | | See Materials and methods; *Figure 4* |
| Recombinant DNA reagent (plasmid) | pET28 PP1-(SG)9-spectrin | This paper | | See Materials and methods; *Figure 4* |
| Recombinant DNA reagent (plasmid) | pET28 PP1-Phactr1 fusion | This paper | | See Materials and methods; *Figure 6* |
| Recombinant DNA reagent | pGro7 plasmid | Takara | 3340 | |
| Sequence-based reagent | Oligonucleotides | This paper | | See Materials and methods; *Supplementary file 4* |
| Sequence-based reagent | Peptides | This paper | | See Materials and methods; *Supplementary file 4* |
| Commercial assay or kit | Q5 Site-Directed Mutagenesis Kit | NEB | e0552s | |
| Commercial assay or kit | NEBuilder HiFi DNA Assembly Cloning Kit | NEB | e5520s | |
| Commercial assay or kit | TMT10plex Isobaric Label Reagent Set, 0.8mg | Thermo | 90111 | |
| Commercial assay or kit | High-Select Fe-NTA Phosphopeptide Enrichment Kit | Thermo | A32992 | |
| Commercial assay or kit | High-Select TiO2 Phosphopeptide Enrichment Kit | Thermo | A32993 | |
| Commercial assay or kit | High pH Reversed Phase Fractionation Kit | Pierce | 84868 | |
| Commercial assay or kit | GenElute mammalian total RNA kit | Sigma | RTN350-1KT | |
| Commercial assay or kit | Transcriptor First Strand cDNA Synthesis kit | Roche | 04897030001 | |
| Commercial assay or kit | SYBR green RT-qPCR mix | Life Technologies | A25742 | |
| Commercial assay or kit | Biomol Green reagent | Enzo Life Sciences | BML-AK111-1000 | |
| Chemical compound | doxycycline | Sigma | D9891-100G | 1 µg/ml |
| Chemical compound | cytochalasin D | Calbiochem | A25742 | |
| Chemical compound | latrunculin B | VWR | 428020-5MG | |

*Continued on next page*

*Appendix 1—key resources table continued*

| Reagent type (species) or resource | Designation | Source or reference | Identifiers | Additional information |
|---|---|---|---|---|
| Chemical compound | Lipofectamine 2000 | Invitrogen | 11668-019 | |
| Chemical compound | Texas Red-phalloidin | Invitrogen | T7471 | |
| Chemical compound | pNPP | Enzo | SV-30770-02 | |
| Chemical compound | SuperSignal West Pico Plus reagent | Thermo | 34577 | |
| Software | Octet software version 7.0 | ForteBio | https://www.fortebio.com/ | |
| Software | Xcalibur | Thermo | https://www.thermofisher.com/order/catalog/product/OPTON-30965 | |
| Software | MaxQuant | *Cox and Mann, 2008* | http://coxdocs.org/doku.php?id=maxquant:start | |
| Software | Perseus | *Tyanova et al., 2016* | https://www.maxquant.org/perseus/ | |
| Software | Weblogo | University of California, Berkeley | https://weblogo.berkeley.edu/logo.cgi/ | |
| Software | GraphPad Prizm | GraphPad | https://www.graphpad.com/scientific-software/prism/ | |
| Software | Image Studio Lite 5.2 | LI-COR | https://www.licor.com/bio/image-studio-lite/ | |
| Software | SnapGene software | Insightful Science | snapgene.com | |
| Chemical compound | Manganese Chloride | Fluka | 221279-500G | |
| Chemical compound | Arabinose | Biosynth limited | MA02043 | |
| Chemical compound | IPTG | Neo Biotech | NB-45-00030-25G | |
| Chemical compound | Chloramphenicol | Acros organic | 227920250 | |
| Chemical compound | Tris | SDS | 10708976001 | |
| Chemical compound | Imidazole | Sigma-Aldrich | I2399-100G | |
| Chemical compound | Sodium Chloride | Sigma Aldrich | S9888-1KG | |
| Chemical compound | Triton X100 | Sigma Aldrich | X100-100ML | |
| Chemical compound | TCEP | Fluorochem | M02624 | |
| Chemical compound | AEBSF | Melford | A20010-5.0 | |
| Chemical compound | Benzamidine | Melford | B4101 | |

*Continued on next page*

*Appendix 1—key resources table continued*

| Reagent type (species) or resource | Designation | Source or reference | Identifiers | Additional information |
|---|---|---|---|---|
| Chemical compound | Complete EDTA Free Protease Inhibitor tablet | Roche | 05056489001 | |
| Chemical compound | Glutathione Sephanrose 4B | GE Healthcare | 17-0756-05 | |
| Chemical compound | Ni-NTA Agarose | Qiagen | 30230 | |
| Chemical compound | Tween 20 | Sigma-Aldrich | P1379-100ML | |
| Chemical compound | BSA | Sigma-Aldrich | A2153-100G | |
| Chemical compound | Lithium Chloride | Hampton research | HR2-631 | |
| Chemical compound | Tri Sodium Citrate | Hampton research | HR2-549 | |
| Chemical compound | PEG 6000 | Hampton research | HR2-533 | |
| Chemical compound | PEG 3350 | Hampton research | HR2-527 | |
| Chemical compound | Sodium Bromide | Hampton research | HR2-699 | |
| Chemical compound | Potassium citrate | Hampton research | HR2-683 | |
| Chemical compound | Bis-Tris-Propane | Sigma-Aldrich | B6755-500G | |
| Chemical compound | Sodium Iodide | Sigma-Aldrich | 383112-100G | |
| Chemical compound | Glycerol | SDS | G7893-2L | |
| Chemical compound | Ethylene Glycol | Sigma-Aldrich | 324558-1L | |
| Other | Amino-PEG500-UC540 membrane | Intavis | | Peptide array membrane; see Materials and methods, and *Figure 5* |

