## [Decision Letter]

**Acceptance summary:**

Your finding that a bound PP1 docking subunit is able to confer primary sequence specificity for PP1 target dephosphorylation is an important advance in our understanding of how the relatively nonspecific PP1 phosphatase catalytic subunit can acquire phosphoprotein target site specificity.

**Decision letter after peer review:**

Thank you for submitting your article "Molecular basis for substrate specificity of the Phactr1/PP1 phosphatase holoenzyme" for consideration by *eLife*. Your article has been reviewed by three peer reviewers, including Tony Hunter as the Reviewing Editor and Reviewer #1, and the evaluation has been overseen by Jonathan Cooper as the Senior Editor. The following individual involved in review of your submission has agreed to reveal their identity: David Barford (Reviewer #2).

The reviewers have discussed the reviews with one another and the Reviewing Editor has drafted this decision to help you prepare a revised submission.

All the reviewers were very positive about the paper and recommend publication in *eLife*. The finding that a bound PP1 accessory subunit is able to confer primary sequence specificity for PP1 target dephosphorylation is an important advance in our understanding of how the nonspecific PP1 catalytic subunit can acquire substrate specificity.

Reviewer #1:

This is an excellent paper describing high resolution crystal structures of a complex between Phactr1, a G-ctin regulated neuronal PP1 activator, and the pSer/Thr protein phosphatase 1 without or with a bound peptide substrate, which provide important new insights into how the Phactr1 subunit provides substrate specificity to the PP1 phosphatase. To understand how Phactr1 provides substrate specificity to PP1, they solved crystal structures of a relevant fragment of Phactr1 bound to PP1c, which showed that in the holoenzyme complex the binding of Phactr1 created a long combinatorial hydrophobic groove that could present an extended target pSer/Thr peptide sequence to the active site, which also contained two Mn^2+^ ions and a free phosphate. The structures revealed four separate motif contacts between the Phactr1 fragment and the PP1 catalytic subunit – RVxF-φφ-R-W. To identify new Phactr1/PP1-specific substrates, they used phosphoproteomic analysis of NIH 3T3 cells inducibly expressing the Phactr1xxx PP1 activating mutant and also compared WT and Phactr1 KO mouse neurons to identify neuronal specific substrates for Phactr1/PP1, identifying a preponderance of substrate proteins involved in regulating the actin cytoskeleton, three of which IRSp53 pS455, spectrin αII pS1031 and afadin pS1275 were validated by making phosphospecific antibodies. This Phactr1/PP1 substrate phosphosite survey also allowed them to define primary sequence preferences. To learn exactly how target phosphopeptide substrates bind to this combinatorial groove, they solved structures of Phactr1/PP1 with a bound phosphatase product peptide, derived from either IRSp53(449-465) or spectrin αII (1025-1039), tethered via a linker to PP1(7-304), which was in turn bound to the Phactr1 fragment; these structures revealed that the substrate peptides were bound into the active site together with a free phosphate molecule. In these these structures the phosphate was bound an inverted fashion compared to the phosphate in the holoenzyme structure and therefore represent enzyme-product complexes. The structures show how the extended binding groove provides primary phosphopeptide sequence specificity. A number of mutagenesis studies were carried out to establish the importance of the observed interactions involved in Phactr1 and substrate binding to PP1. They also solved another PP1-Phactr1 structure with a Glu455 phosphomimetic IRSp53 substrate peptide linked to PP1, and interestingly found that the Glu455 was not docked into the active site, and instead the peptide had shifted one register such that Thr546 and a phosphate ion were docked there, consistent with glutamate being a poor phosphomimetic.

The finding that a bound PP1 docking subunit is able to confer primary sequence specificity for PP1 target dephosphorylation is an important advance in our understanding of how the nonspecific PP1 catalytic subunit can acquire substrate specificity. This is a very comprehensive study and I have no substantive comments.

Reviewer #2:

This manuscript by Treisman and colleagues explores the molecular basis for the specificity of the Ser/Thr PP-family phosphatase PP1. The question of whether PP1 holoenzymes possess intrinsic substrate specificity per se, and if so, what is the molecular basis for this specificity, have been long-standing important questions. In this respect the, PPP phosphatases in particular, and the whole phosphatase family in general, are much less well understood than the protein kinases. This study is a tour de force, addressing these questions definitively, thoroughly and vigorously. It represents a landmark study for the field. Previous work had defined how the PP1 catalytic subunit (PP1c) interacts with short sequence motifs to target PP1c to regulatory subunits and substrates, thereby providing some degree of specificity at the level of substrate and cell localisation. However, it was unknown how substrates bind to PP1, or how regulatory subunits/cofactors contribute to, and modulate the structure of the substrate-binding site on the PP1 holoenzyme.

The authors first determined the crystal structure of PP1 in complex with the PP1-region of Phactr1, a neuronally-enriched PP1 cofactor, controlled by G-actin. This revealed that Phactr1 binds PP1 via multiple motifs including the RVxF, a phi-phi motif, a novel Trp motif and Arg-motif, the authors describe this as an RVxF-phi-phi-R-W string. Additionally, a 20-residue amphipathic helix, contacts PP1. These interactions remodel PP1's hydrophobic groove and create a new composite binding site adjacent to the catalytic site. Leu519 of the RVxF motif also contacts G-actin, explaining how G-actin binding to Phactr proteins regulates PP1 activity. The composite active site has a pronounced basic rim. The role of many of key-interacting residues were validated by mutagenesis/affinity studies.

The authors used an activated derivative of Phactr1 that constitutively binds and activates PP1 in cells. This allowed them to identify PP1- Phactr1 substrates. Interestingly, these substrates show a defined consensus dephosphorylation sequence: basic residues N-terminal to the pSer/Thr, followed by acidic residues and aliphatics (Leu) at P4 and P5. Very high-resolution crystal structures of peptides modelled on the dephosphorylation sequences of IRSp53 and spectrin (fused to PP1c) were determined to obtain a structural rationale for the substrate specificity of PP1-Phactr1 complexes. These structures elegantly explain the preference for acidic residues C-terminal to pSer/Thr through contacts to the basic rim of Phactr1, and how the Leu residues at +p4/p5 interact with a hydrophobic pocket formed from both PP1c and Phactr1. Unexpectedly the diLeu motif of the substrates can be shifted in register by one residue relative to pSer/Thr to bind at the hydrophobic pocket. This illustrates a degree of plasticity in the recognition of substrates by PP1-Phactr1 complexes.

As a bit of a bonus, the authors also trapped a phosphate at the catalytic site in their Phactr1/PP1-IRSp53/spectrin complexes. Interestingly the geometry of the phosphate is inverted relative to the holoenzyme (without substrate). The later geometry is similar to that found in earlier studies of PP1c, which likely represents how the phosphate of an unphosphorylated pSer/Thr would bind. Thus, the authors have fortuitously captured the dephosphorylated product showing the phosphate in an inverted configuration. The extremely high resolution of their structures allows definitive location of oxygen atoms, so this work definitively supports the model of PP1 catalysis proposed some 25 years ago.

All the crystal structures are of exceptional quality, with one complex determined at 1.09 A resolution, – likely by far the highest for any phosphatase (and the vast majority of crystal structures).

Overall this is an excellent and very interesting study that warrants publication in *eLife*.

1) The only scientific question is the MSA in Figure 1—figure supplement 1A. The authors have aligned Phactr paralogs with an insertion in Phactr2 and Phactr3 in the “V” position of the “RVxF” motif. This is extremely unlikely, because the Val residue is critical to the motif. A structure-based alignment would align Leu of the “IL” preceding “RF” with the valine. This could be tested experimentally, but is not a condition for publishing this paper.

Reviewer #3:

This is a beautiful combination of structural biology and phosphoproteomics that interact to delineate the mechanism of action of a PP1 targeting subunit. They identify how Phactr1 binds to PP1, a structure supported by diverse mutagenesis studies. Phosphoproteomics identifies a number of substrates that make biological sense, and they validate a handful with phospho-epitope antibodies. They are able to crystalize two substrate peptides in the Phactr1-PP1 complex using a tether, which raises issues of forced changes in orientation. However, the peptide interactions are supported mutagenesis studies and high similarity of interactions between the two independent peptides.

The next step would be to tie the changes in phosphorylation at specific sites identified to cellular phenotypes, something not achieved here. But that is not to detract from a very ambitious and informative manuscript that will be of broad interest to those who think about how phosphorylation is controlled, and how phosphatases find their substrates.

---

## [Author Response]

Reviewer #2:[…]Overall this is an excellent and very interesting study that warrants publication in eLife.1) The only scientific question is the MSA in Figure 1—figure supplement 1A. The authors have aligned Phactr paralogs with an insertion in Phactr2 and Phactr3 in the “V” position of the “RVxF” motif. This is extremely unlikely, because the Val residue is critical to the motif. A structure-based alignment would align Leu of the “IL” preceding “RF” with the valine. This could be tested experimentally, but is not a condition for publishing this paper.

The alignment is shown to emphasise the sequence relationships, and is not structure-based, which as the reviewer points out raises a problem. To address this the Phactr1 RVxF-φφ-R-W PP1 contact residues are highlighted in blue, as are the presumed structurally equivalent residues residues on the Phactr2, 3 and 4 sequences. This is explained in the legend of Figure 1—figure supplement 1A.